# Aire-dependent genes undergo Clp1-mediated 3'UTR shortening associated with higher transcript stability in the thymus

Clotilde Guyon[1†], Nada Jmari[1†], Francine Padonou[1,2], Yen-Chin Li[1], Olga Ucar[3], Noriyuki Fujikado[4‡], Fanny Coulpier[5], Christophe Blanchet[6], David E Root[7], Matthieu Giraud[1,2]*

[1]Institut Cochin, INSERM U1016, Université Paris Descartes, Sorbonne Paris Cité, Paris, France; [2]Université de Nantes, Inserm, Centre de Recherche en Transplantation et Immunologie, UMR 1064, ITUN, F-44000, Nantes, France; [3]Division of Developmental Immunology, German Cancer Research Center, Heidelberg, Germany; [4]Division of Immunology, Department of Microbiology and Immunobiology, Harvard Medical School, Boston, United States; [5]Ecole Normale Supérieure, PSL Research University, CNRS, INSERM, Institut de Biologie de l'Ecole Normale Supérieure (IBENS), Plateforme Génomique, Paris, France; [6]Institut Français de Bioinformatique, IFB-Core, CNRS UMS 3601, Evry, France; [7]The Broad Institute of MIT and Harvard, Cambridge, United States

**\*For correspondence:**
matthieu.giraud@inserm.fr

[†]These authors contributed equally to this work

**Present address:** [‡]Lilly Biotechnology Center, Lilly Research Laboratories, Eli Lilly and Company, San Diego, United States

**Competing interests:** The authors declare that no competing interests exist.

**Abstract** The ability of the immune system to avoid autoimmune disease relies on tolerization of thymocytes to self-antigens whose expression and presentation by thymic medullary epithelial cells (mTECs) is controlled predominantly by Aire at the transcriptional level and possibly regulated at other unrecognized levels. Aire-sensitive gene expression is influenced by several molecular factors, some of which belong to the 3'end processing complex, suggesting they might impact transcript stability and levels through an effect on 3'UTR shortening. We discovered that Aire-sensitive genes display a pronounced preference for short-3'UTR transcript isoforms in mTECs, a feature preceding Aire's expression and correlated with the preferential selection of proximal polyA sites by the 3'end processing complex. Through an RNAi screen and generation of a lentigenic mouse, we found that one factor, Clp1, promotes 3'UTR shortening associated with higher transcript stability and expression of Aire-sensitive genes, revealing a post-transcriptional level of control of Aire-activated expression in mTECs.

## Introduction

Immunological tolerance is a key feature of the immune system that protects against autoimmune disease by preventing immune reactions against self-constituents. Central tolerance is shaped in the thymus and relies on a unique property of a subset of medullary thymic epithelial cells (mTECs). This subset is composed of mTEChi that express high levels of MHC class II molecules and a huge diversity of self-antigens (*Danan-Gotthold et al., 2016*; *Kyewski and Klein, 2006*). Thus, developing T lymphocytes in the thymus are exposed to a broad spectrum of self-antigens displayed by mTEChi. Those lymphocytes that recognize their cognate antigens undergo either negative selection, thereby preventing the escape of potentially harmful autoreactive lymphocytes out of the thymus, or differentiate into thymic regulatory T cells beneficial for limiting autoreactivity (*Cowan et al., 2013*;

*Goodnow et al., 2005*; *Klein et al., 2014*). Self-antigens expressed by mTEChi include a large number of tissue-restricted antigens (TRAs), so-named because they are normally restricted to one or a few peripheral tissues (*Derbinski et al., 2001*; *Sansom et al., 2014*). A large fraction of these TRAs in mTEChi are induced by a single transcriptional activator that is expressed almost exclusively in these cells - the autoimmune regulator Aire. Mice deficient for the *Aire* gene exhibit impaired TRA expression in mTEChi, whereas TRA expression remains normal in peripheral tissues of these mice. Consistent with inadequate development of central tolerance, *Aire* knockout (KO) mice develop autoantibodies directed at some of these TRAs, resulting in immune infiltrates in multiple tissues (*Anderson et al., 2002*). Correspondingly, loss-of-function mutations in the human *AIRE* gene result in a multi-organ autoimmune disorder known as autoimmune polyglandular syndrome type 1 (*Nagamine et al., 1997*; *Peterson et al., 2004*).

How the expression of thousands of Aire-sensitive self-antigen genes is controlled in mTEChi has been a subject of extensive investigation. Significant progress has been made, notably through the identification of a number of molecular factors that further activate the expression of prototypic Aire-sensitive genes in a model employing cell lines that express Aire ectopically by transfection with a constitutive *Aire* expression vector (*Abramson et al., 2010*; *Giraud et al., 2014*). These studies revealed a role for relaxation of chromatin in front of the elongating RNA polymerase (RNAP) II by the PRKDC-PARP1-SUPT16H complex (*Abramson et al., 2010*) and for an HNRNPL-associated release of the stalled RNAPII (*Giraud et al., 2014*). However, the effect of most of the identified factors on the full set of Aire-sensitive genes in mTEChi is unknown. It remains also uncertain whether these factors partake in a molecular mechanism directly orchestrated by Aire or in a basal transcriptional machinery that would control the expression of Aire-sensitive genes even before Aire is expressed in mTEChi. Lack of knowledge of the precise modus operandi of the identified factors potentially leaves major aspects of promiscuous mTEChi gene expression unknown.

Among the identified factors, seven of them, namely CLP1, DDX5, DDX17, PABPC1, PRKDC, SUPT16H and PARP1, have been reported to belong to the large multi-subunit 3′ end processing complex (*de Vries et al., 2000*; *Shi et al., 2009*) which controls pre-mRNA cleavage and polyadenylation at polyA sites (pAs) (*Colgan and Manley, 1997*). Hence, we asked whether any of these identified factors could influence Aire-sensitive gene expression partially or entirely by the way of an effect on pre-mRNA 3′ end maturation. Deep sequencing approaches have revealed that the vast majority of protein-coding genes in mammal genomes (70–79%) have multiple pAs mostly located in 3′UTRs (*Derti et al., 2012*; *Hoque et al., 2013*). These genes are subject to differential pA usage through alternative cleavage and polyadenylation directed by the 3′ end processing complex and are transcribed as isoforms with longer or shorter 3′UTRs depending on the pA usage (*Tian and Manley, 2013*). The 3′ end processing complex is composed of a core effector sub-complex comprising CLP1 (*de Vries et al., 2000*; *Mandel et al., 2008*) and a number of accessory proteins that include DDX5, DDX17, PABPC1, PRKDC, SUPT16H and PARP1 (*Shi et al., 2009*). Although the individual roles of accessory proteins on differential pA usage remain largely unknown, the core protein Clp1 has been reported to favor proximal pA selection in yeast based on depletion experiments (*Holbein et al., 2011*). Similarly, increasing levels of Clp1 bound to the 3′ end processing complex was also shown to favor proximal pA selection and shorter 3′UTR isoforms (*Johnson et al., 2011*). In contrast, a preference for distal pA selection was reported for higher Clp1 levels in a mouse myoblast cell line based on siRNA Clp1 loss-of-function experiments (*Li et al., 2015*).

Given the observations from prior work that many of the genes, other than Aire itself, that modulate the expression of Aire-induced genes, are members of the 3′ end processing complex, and that one such member of this complex, Clp1, has been reported to affect pA selection, we speculated that 3′UTR length and regulation might be involved in expression of TRAs in mTEChi. We therefore set out to investigate relationships between Aire sensitivity, 3′ end processing, and pA selection in mTEChi.

## Results

### Aire-sensitive genes show a preference for short-3'UTR transcript isoforms in mTEChi

To assess the proportion of long and short-3'UTR transcript isoforms in mTEChi, we selected the genes that harbor potential proximal alternative pAs in their annotated 3'UTRs according to the Pol-yA_DB 2 database which reports pAs identified from comparisons of cDNAs and ESTs from a very large panel of peripheral tissues (*Lee et al., 2007*; *Figure 1—source data 1*). For each gene, the relative expression of the long 3'UTR isoform versus all isoforms could be defined as the distal 3'UTR (d3'UTR) ratio, that is the expression of the region downstream of the proximal pA (d3'UTR) normalized to the upstream region in the last exon (*Figure 1A*). To determine whether the Aire-sensitive genes exhibit a biased proportion of long and short-3'UTR isoforms in mTEChi, we first performed RNA deep-sequencing (RNA-seq) of mTEChi sorted from WT and *Aire*-KO mice in order to identify the Aire-sensitive genes, that is those upregulated by Aire (*Figure 1B* and *Figure 1—source data 2*). We then compared the distribution of d3'UTR ratios in Aire-sensitive versus Aire-neutral genes in WT mTEChi. We found a significant shift towards smaller ratios, revealing a preference of Aire-sensitive genes for short-3'UTR isoforms in mTEChi (*Figure 1C*, *Figure 1—figure supplement 1A* and *Figure 1—source data 3* and *4*). The Aire-sensitive genes are expressed at much lower levels than the Aire-neutral genes. Hence, we sought to distinguish whether the observed smaller d3'UTR ratios were specifically associated with Aire-sensitive genes or not simply a co-correlate of low expression. We examined the d3'UTR ratios across expression levels of both Aire-sensitive and neutral genes. While the d3'UTR ratios vary dramatically across genes, at all expression levels, the loess-fitted curve of the Aire-sensitive genes is significantly lower than the one of the Aire-neutral genes, therefore revealing a preference of Aire-sensitive genes for smaller d3'UTR ratios in mTEChi that is independent of the levels of gene expression (*Figure 1—figure supplement 2A*). Since a much larger proportion of Aire-sensitive genes than Aire-neutral genes are known to be TRA genes, we further asked whether the preference of genes for short-3'UTRs was more aligned with Aire-sensitivity or being a TRA gene. To this end, we compared the d3'UTR ratios between the TRA and non-TRA genes as defined in reference (*Sansom et al., 2014*) in the subsets of Aire-sensitive and neutral genes. In these mTEChi, the short-3'UTR isoform preference was observed preferentially in Aire-sensitive genes regardless of whether or not they were TRA genes (*Figure 1D*).

To discriminate whether the preference for short-3'UTR isoforms in the Aire-sensitive genes was directly associated with the process of Aire's induction of gene expression or rather was a feature of Aire-sensitive genes preserved in the absence of Aire, we analyzed the proportion of long 3'UTR isoforms for the Aire-sensitive genes in *Aire*-KO mTEChi. We note that by definition these genes are all expressed at lower levels in the absence of Aire but that most are still expressed at levels sufficient to determine 3'UTR isoform ratios. We observed a preference for the smaller ratios (*Figure 1E*, *Figure 1—figure supplement 1B* and *Figure 1—source data 3* and *4*) in Aire-sensitive versus neutral genes in the *Aire*-KO cells. For these KO mTEChi, we once again examined the d3'UTR ratios across all expression levels of both Aire-sensitive and neutral genes and again, like in WT mTEChi, found that the preference of Aire-sensitive genes for smaller d3'UTR ratios was independent of the levels of gene expression (*Figure 1—figure supplement 2B*). We further noted that the majority (~90%) of Aire-sensitive genes that exhibited small d3'UTR ratios (<0.25) in WT mTEChi were also characterized by small d3'UTR ratios in *Aire*-KO mTEChi (*Figure 1F* and *Figure 1—figure supplement 1C*), Together, these observations showed that the short-3'UTR isoform preference of Aire-sensitive genes in mTEChi was specific to those genes responsive to Aire, whether or not Aire was actually present.

To determine whether the genes sensitive to Aire exhibit shorter 3'UTR isoforms in mTEChi than in their normal tissue of expression, we set out to identify the Aire-sensitive genes showing tissue-specific or selective expression and to calculate the d3'UTR ratios of the identified Aire-sensitive TRA genes in their tissue(s) of expression. To this end, we collected RNA-seq datasets corresponding to a variety of tissues (*Shen et al., 2012*; *van den Berghe et al., 2013*; *Warren et al., 2013*) and selected in each tissue the set of Aire-sensitive genes for which we identified a restricted expression in comparison to the other tissues by applying the SPM (Specificity Measurement) method (*Pan et al., 2013*; *Figure 1—figure supplement 1D,E*). We then calculated the d3'UTR ratios of these genes in each peripheral tissue and found variable d3'UTR ratios across tissues with the

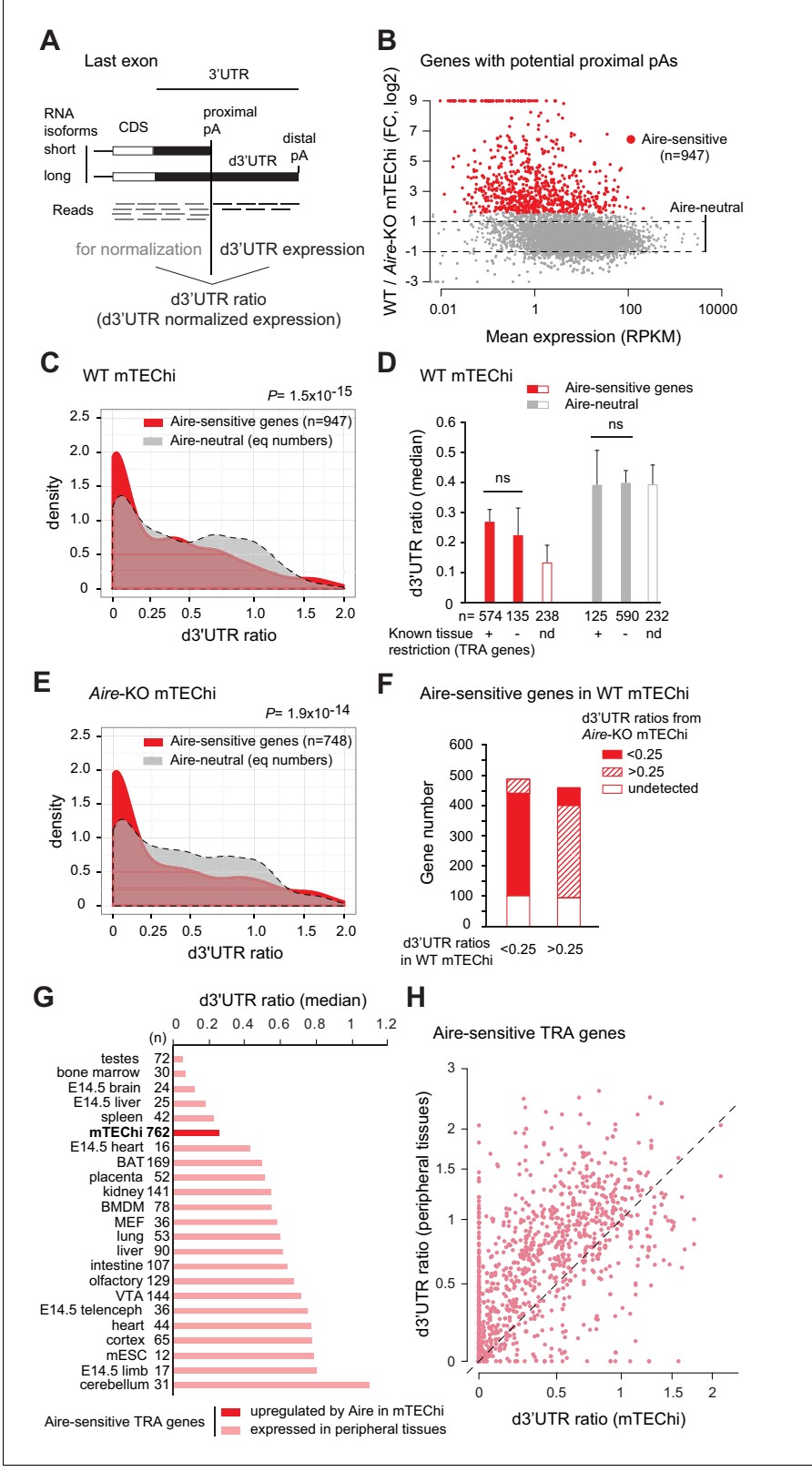

**Figure 1.** Preference of Aire-sensitive genes for short-3'UTR transcript isoforms in mTEChi. (**A**) Schematic of pA usage and 3'UTR isoform expression. 3' ends of RNA isoforms of a hypothetical gene are shown. Usage of the proximal pA results in a reduced proportion of the long 3'UTR isoform, estimated by the d3'UTR ratio. (**B**) RNA-seq differential expression (fold-change) between WT and *Aire*-KO mTEChi sorted from a pool of 4 thymi. Red

*Figure 1 continued*

dots show genes upregulated by threefold or more (Z-score criterion of p<0.01) (Aire-sensitive). Genes between the dashed lines have a change in expression less than twofold (Aire-neutral). (C) Densities of d3'UTR ratios of Aire-sensitive genes upregulated by Aire in mTEChi and of Aire-neutral genes; equal number (n = 947) of neutral genes included, asinh scale. (D) Median of d3'UTR ratios of Aire-sensitive and neutral genes depending on whether their peripheral expression is tissue-restricted, or not. Genes whose classification is not established are called 'not determined' (nd) and represented by an open box. (E) Densities of d3'UTR ratios of Aire-sensitive and neutral genes in *Aire*-KO mTEChi; equal number (n = 748) of neutral genes included, asinh scale. (F) Proportion of Aire-sensitive genes with d3'UTR ratios < 0.25 or>0.25 in *Aire*-KO mTEChi among those with d3'UTR ratios < 0.25 or>0.25 in WT mTEChi. (G) Median of d3'UTR ratios calculated from RNA-seq data for Aire-sensitive genes with tissue-restricted expression in mTEChi and 22 mouse tissues. Duplicate reads were discarded to allow more accurate dataset comparison. Cell types were arranged in ascending order based on the median of the d3'UTR ratios of their Aire-sensitive TRA genes. (H) Scatterplot of d3'UTR ratios of 762 TRA genes in mTEChi and their respective tissue of expression.

The online version of this article includes the following source data and figure supplement(s) for figure 1:

**Source data 1.** d3'UTR annotation files in mice and humans for RNA-seq and microarray analyses.
**Source data 2.** Annotation files in mice and humans for RNA-seq differential gene expression.
**Source data 3.** d3'UTR ratio calculation in WT and *Aire*-KO mTEChi.
**Source data 4.** List of Aire-sensitive genes with proximal pAs in mTEChi.
**Figure supplement 1.** Validation and examples of the preferred short-3'UTR isoform expression of Aire-sensitive genes in mTEChi.
**Figure supplement 2.** Scatterplot of d3'UTR ratio versus gene expression.
**Figure supplement 3.** Impact of sequencing depth and read length on d3'UTR ratios.

mTEChi result falling in the low end of this range (*Figure 1G* and *Figure 1—figure supplement 1F*). Taking the Aire-sensitive TRA genes individually, we compared their d3'UTR ratios in mTEChi versus their respective tissue of expression and found a dramatic bias towards higher ratios in peripheral tissues, confirming the preference of Aire-sensitive TRA genes for short-3'UTR transcript isoforms in mTEChi, as compared to the periphery (*Figure 1H*). In addition, we confirmed that small d3'UTR ratios of Aire-sensitive genes in mTEChi were not correlated with the sequencing depth of the mTE-Chi RNA-seq libraries nor with the length of the generated reads (*Figure 1—figure supplement 3*). These findings show that the Aire-sensitive TRA genes exhibit an increased proportion of short-3'UTR transcript isoforms in mTEChi versus a majority of peripheral tissues.

## The 3' end processing complex is preferentially located at proximal pAs of Aire-sensitive genes in AIRE-negative HEK293 cells

We sought to determine whether the preference for short-3'UTR isoforms of Aire-sensitive genes observed in *Aire*-KO mTEChi and conserved upon upregulation by Aire, is associated with a preferred proximal pA usage driven by the 3' end processing complex. Current techniques dedicated to localize RNA-binding proteins on pre-mRNAs, for example ultraviolet crosslinking and immunoprecipitation (CLIP)-seq (*König et al., 2012*), need several millions of cells, precluding their use with primary *Aire*-KO or WT mTEChi for which only ~30,000 cells can be isolated per mouse. To circumvent this issue, we used the HEK293 cell line for these experiments. HEK293 cells are (i) negative for AIRE expression, (ii) responsive to the transactivation activity of transfected Aire (*Abramson et al., 2010*; *Giraud et al., 2012*), and (iii) have been profiled for the RNA binding of the 3' end processing components by Martin et al. by PAR-CLIP experiments (*Martin et al., 2012*). Similarly to what we found in WT and *Aire*-KO mTEChi, we observed in *Aire*-transfected and Ctr-transfected HEK293 cells significant lower d3'UTR ratios for genes identified by *Aire* transfection to be Aire-sensitive versus Aire-neutral genes (*Figure 2A* and *Figure 2—figure supplement 1A*). It should be noted that many Aire-neutral genes featured moderately lower d3'UTR ratios in *Aire*-transfected HEK293 cells than Ctr-transfected cells, as a possible effect of Aire itself on an overall 3'UTR shortening in these cells, but this did not produce the large proportion of very small d3'UTR ratios (<0.2) that were exhibited by the Aire-sensitive genes in *Aire* or Ctr-transfected HEK293 cells. Within HEK293 cells, localization of the 3'end processing complex at proximal or distal pAs was performed by analyzing the binding pattern of CSTF2, the member of the core 3' end processing complex that has been reported to exhibit the highest binding affinity for the maturing transcripts at their cleavage sites close to pAs

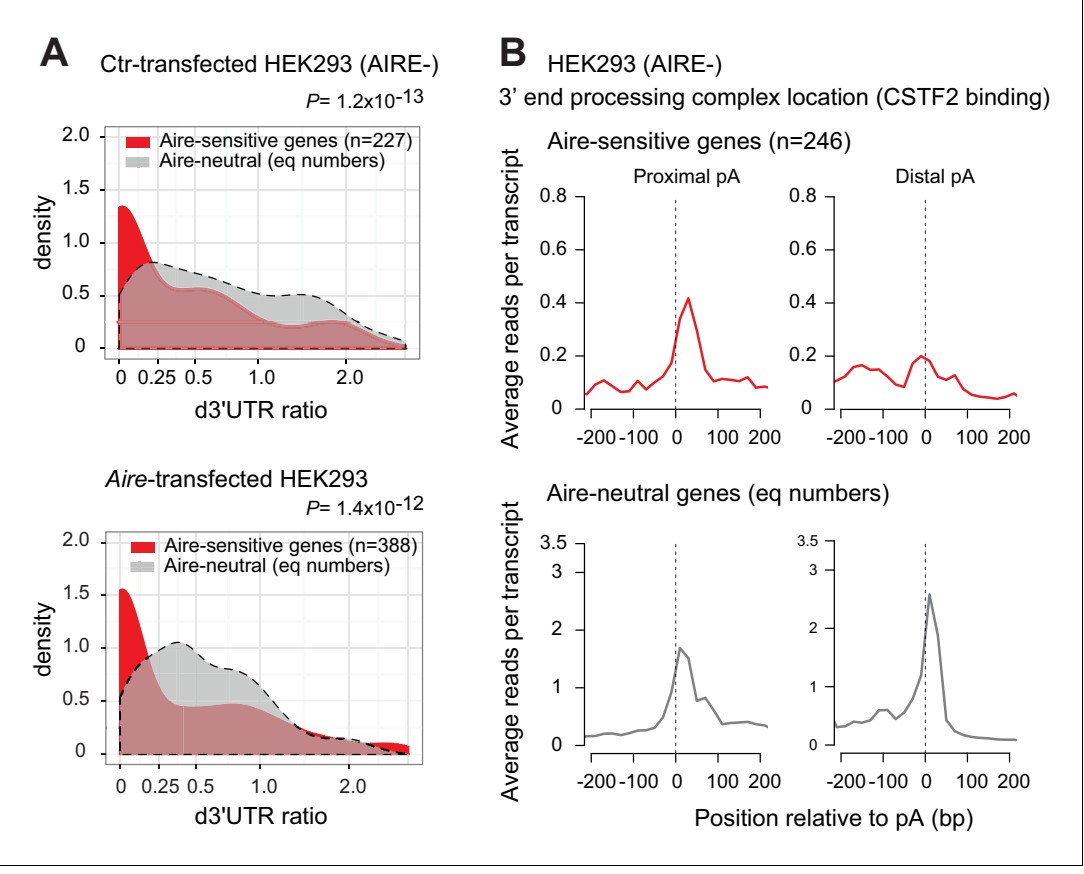

**Figure 2.** Increased binding of the 3'end processing complex at proximal pAs of Aire-sensitive genes in HEK293 cells. (**A**) Densities in Ctr-transfected HEK293 cells (AIRE-) (*Top*) and *Aire*-transfected HEK293 cells (*Bottom*) of d3'UTR ratios from RNA-seq data of Aire-sensitive and neutral genes identified after *Aire* transfection; equal numbers of neutral genes included (n = 227 and n = 388, respectively), asinh scale. (**B**) Average density of reads from PAR-CLIP analyses in HEK293 cells (AIRE-) of CSTF2 protein as a marker of the 3' end processing complex, in the vicinity of proximal and distal pAs of Aire-sensitive and neutral genes. Equal number (n = 246) of neutral genes included.

The online version of this article includes the following source data and figure supplement(s) for figure 2:

**Source data 1.** Genomic location of pAs on hg19 extracted from the PolyA_DB 2 database for CLIP-seq analysis.
**Figure supplement 1.** Correlation of the binding of the 3' end processing complex with proximal pA location.

(*Martin et al., 2012*). We first validated for Aire-neutral genes that lower d3'UTR ratios correlated with the preferential location of the 3' end processing complex at proximal pAs (*Figure 2—figure supplement 1B*). Then we compared the location of the complex between the Aire-sensitive and neutral genes, and found a dramatic preference for proximal pAs versus distal pAs at Aire-sensitive genes (*Figure 2B*), showing that the 3' end processing complex was already in place at proximal pAs of Aire-sensitive genes before Aire expression was enforced in HEK293 cells.

## CLP1 promotes 3'UTR shortening and higher expression at Aire-sensitive genes in HEK293 cells

Since the preference for short-3'UTR isoform expression of Aire-sensitive genes is associated with the increased binding of the 3' end processing complex to proximal pAs in AIRE-negative HEK293 cells, we asked whether factors that belong to the large 3' end processing complex (*de Vries et al., 2000*; *Shi et al., 2009*) could account for the short-3'UTR transcript isoform preference of Aire-sensitive genes in these cells and affect the expression of these genes. Among the members of the core and accessory subunits of the 3' end processing complex, the cleavage factor CLP1 (core) and also DDX5, DDX17, PABPC1, PRKDC, SUPT16H and PARP1 (accessory) have been reported to control

Aire-sensitive gene expression in an Aire-positive context (*Giraud et al., 2014*) with the possibility that the effect of some of these factors could result from their action on the basal expression of Aire-sensitive genes and therefore be observed in the absence of Aire. In addition to these candidate factors, we also tested the effect of HNRNPL, which although not part of the 3' end processing complex, has been shown to regulate 3' end processing of some human pre-mRNAs (*Millevoi and Vagner, 2010*) and control Aire transactivation function in mTEChi (*Giraud et al., 2014*).

First, to determine whether any of the candidate factors could be involved in 3'UTR shortening per se, we carried out short hairpin (sh)RNA-mediated interference in AIRE-negative HEK293 cells and generated expression profiles using Affymetrix HuGene ST1.0 microarrays (*Figure 3—source data 1*). These arrays typically include one or two short probes per exon, and for approximatively 35% of all genes with potential proximal pAs they also include at least two short probes in the d3'UTR region. In spite of the limited d3'UTR coverage and lower accuracy of hybridization measurements based on two short probes only, we found these arrays adequate to detect 3'UTR length variation at the genome-scale level. For each gene with a microarray-detectable d3'UTR region, the d3'UTR isoform ratio was calculated by dividing the measured d3'UTR expression by the whole-transcript expression based on all short probes mapping to the transcript (*Figure 3—source data 1* and *2*). Comparison of the percentages of genes exhibiting a significant increase or decrease of the d3'UTR ratios upon knockdown of a candidate gene versus the control LacZ gene, was used to evaluate the specific impact of the candidate on 3'UTR isoform expression (*Figure 3—source data 3*). For each candidate gene, we measured the knockdown efficiency of, typically, five different shRNAs and selected the three most effective (*Supplementary file 1*). With a threshold of at least two shRNAs per gene producing a significant excess of genes with increased d3'UTR ratios, we found that CLP1, DDX5, DDX17, PARP1 and HNRNPL contributed to 3'UTR shortening in HEK293 cells (*Figure 3A* and *Figure 3—figure supplement 1*). The absence of effect of PRKDC, SUPT16H and PABPC1 could indicate that these factors do not contribute to the activity of 3' end processing subcomplexes or that other factors, perhaps with redundant function, can compensate for their loss. As a control, we also performed knockdown of CPSF6, a core member of the 3' end processing complex, that has been consistently reported to promote general 3'UTR lengthening (*Li et al., 2015*; *Martin et al., 2012*). As expected, knockdown of *CPSF6* revealed a skewed distribution towards decreased d3'UTR ratios (*Figure 3—figure supplement 2*).

Then, we sought among the candidate factors that had an effect on 3'UTR shortening, those that also selectively impacted the basal (in absence of Aire) expression of the Aire-sensitive genes possessing proximal pAs by analyzing microarray whole-transcript expression in control (Ctr) and knockdown HEK293 cells. For each candidate factor, we computed the rank of differential expression of all genes in Ctr versus HEK293 cells knockdown for the tested candidate, and found that *CLP1* was the only gene whose knockdown by two distinct shRNAs led to a significant reduction of the expression of Aire-sensitive genes (*Figure 3B*), focusing our subsequent study on CLP1. The effect of *CLP1* knockdown on Aire-sensitive genes lacking annotated proximal pAs was much less than for the genes with proximal pAs, but a smaller effect on the annotated single-pA genes was observed for one of two CLP1 shRNAs (*Figure 3C*). This might simply be because some genes tagged as "proximal pA-" in the incomplete PolyA_DB 2 database possessed undiscovered proximal pAs in mTEChi and were therefore responsive to the action of CLP1. Our findings regarding the differential response of pA+ and pA- genes to CLP1 loss-of-function suggested that the effect of CLP1 on Aire-sensitive gene expression in HEK293 cells was dependent on the potential of Aire-sensitive genes to switch from using a distal to a proximal pA site.

To assess the effect of CLP1 on 3'UTR shortening at Aire-sensitive genes, we performed RNA-seq experiments in Ctr and *CLP1* knockdown HEK293 cells. Comparison of the 3' end profiles revealed a statistically significant increase of the median d3'UTR ratios of Aire-sensitive genes for both *CLP1* hit shRNAs (*Figure 3D*, *Left*), showing that CLP1 is able to promote 3'UTR shortening at Aire-sensitive genes in HEK293 cells. The d3'UTR ratios of Aire-sensitive genes after *CLP1* knockdown did not reach those of Aire-neutral genes which could simply be due to remaining *CLP1* in these cells following knockdown (measured knockdown as 75% and 72% for shRNA 2 and 3, respectively) or could indicate that additional non-CLP1-dependent factors also contribute to the difference in d3'UTR ratios of Aire-sensitive versus Aire-neutral genes. In contrast to Aire-sensitive genes, only small, statistically insignificant increases of d3'UTR ratios of Aire-neutral genes could be detected in *CLP1* knockdown HEK293 cells, indicating that the effect of CLP1 on 3'UTR shortening preferentially

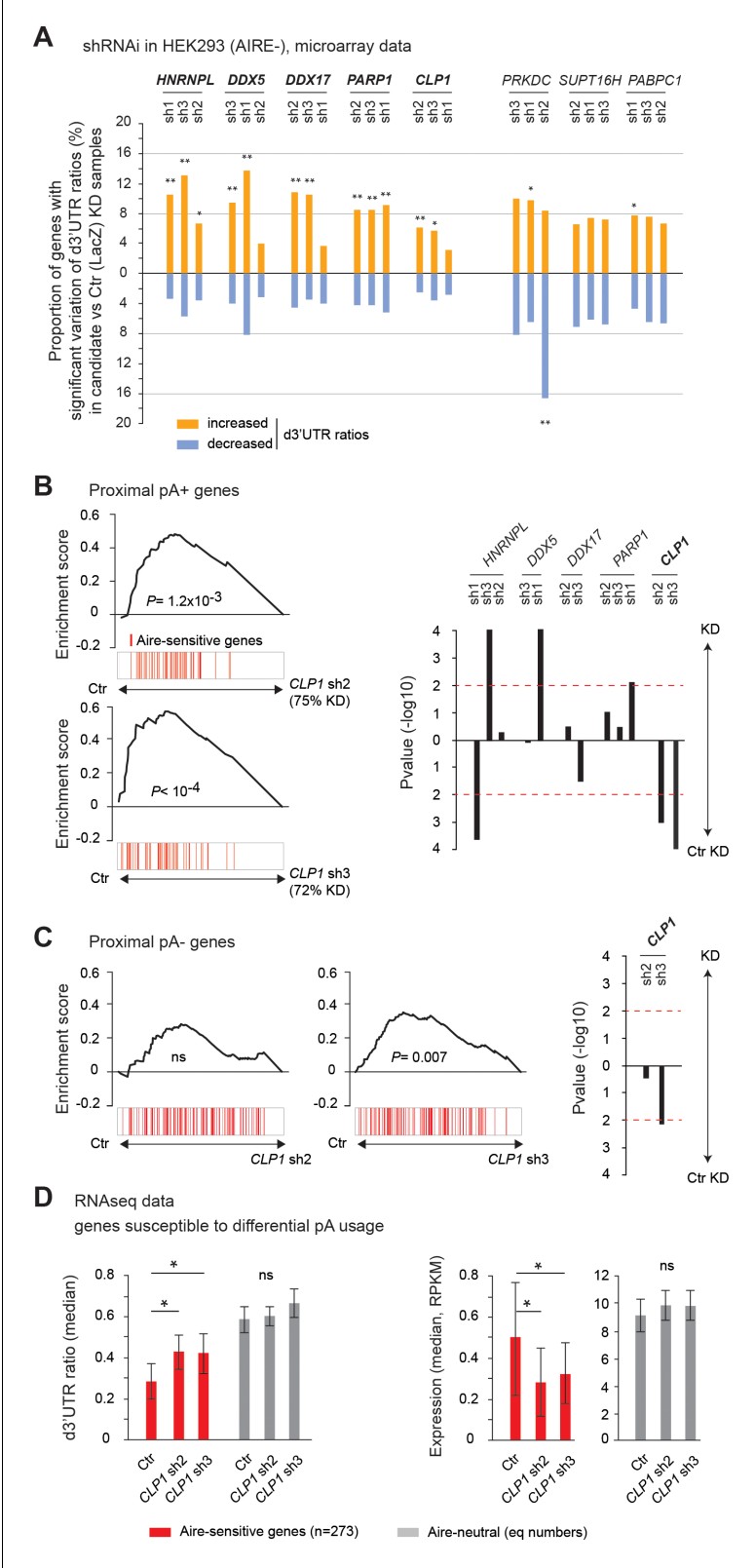

**Figure 3.** CLP1 controls the expression of Aire-sensitive genes with proximal pAs and their shortening in HEK293 cells. (**A**) Individual probe-level analysis of microarray data from knockdown and control HEK293 cells. Vertical bars represent the proportion of genes with a significant increase or decrease of d3'UTR ratios in the candidate vs. Ctr (LacZ) knockdown samples. *p<$10^{-4}$, **p<$10^{-9}$ (Chi-squared test). (**B**), (**C**) Gene Set Enrichment Analysis of Aire-

*Figure 3 continued on next page*

*Figure 3 continued*

sensitive genes among Ctr (LacZ) vs *CLP1* KD ranked expression datasets of all genes susceptible to differential pA usage (with potential proximal pAs: proximal pA+) (**B**) and without potential proximal pAs (proximal pA-) (**C**) in their 3'UTRs as identified in the PolyA_DB 2 database. Significance and direction of the enrichment is shown for each hit shRNA, as well as *P* value thresholds of 0.01 by horizontal red dashed lines (*Right*). (**D**) Median of d3'UTR ratios (*Left*) and expression values (*Right*) from RNA-seq data of Aire-sensitive and neutral genes in HEK293 cells infected by lentiviruses containing *CLP1* hit shRNAs or the Ctr (LacZ) shRNA; equal number (n = 273) of neutral genes included, error bars show the 95% confidence interval of the medians. *p<0.05.

The online version of this article includes the following source data and figure supplement(s) for figure 3:

**Source data 1.** Human Gene ST1.0 microarray probeset and individual probe expression extraction.
**Source data 2.** Microarray individual probe d3'UTR mapping and d3'UTR ratio calculation.
**Source data 3.** d3'UTR ratio imbalance obtained from microarray data.
**Figure supplement 1.** Effect of shRNA-mediated interference of candidate factors on d3'UTR ratios.
**Figure supplement 2.** CPSF6 promotes 3'UTR lengthening.
**Figure supplement 3.** Proportion of genes subject to CLP1-mediated 3'UTR shortening in Aire-sensitive and neutral genes in HEK293 cells.
**Figure supplement 4.** Impact of *CLP1* knockdown on the expression levels of Aire-sensitive genes in *Aire*-transfected HEK293 cells.

---

affects the Aire-sensitive genes in mTEChi. The preferential 3'UTR shortening effect of CLP1 at Aire-sensitive genes was further supported by the observation of an enhanced proportion of genes subject to CLP1-mediated 3'UTR shortening among the Aire-sensitive versus neutral genes in HEK293 cells (*Figure 3—figure supplement 3*). This finding is consistent with the microarray results (*Figure 3A*) indicating that a small proportion of genes in the genome with microarray-detectable d3'UTR regions underwent 3'UTR length variation after *CLP1* knockdown. Finally, using RNA-seq data, we confirmed the effect of *CLP1* knockdown on the selective reduction of the expression of Aire-sensitive genes that were annotated to have alternative proximal pAs in HEK293 cells (*Figure 3D*, *Right*). Furthermore, we observed that the extent of the effect of *CLP1* knockdown on the levels of Aire-sensitive gene expression in *Aire*-transfected HEK293 cells (*Figure 3—figure supplement 4*) was similar, although slightly stronger than the one we described above in AIRE-negative HEK293 cells.

Together these findings showed that CLP1 is able to promote 3'UTR shortening and increase expression of Aire-sensitive genes with proximal pAs, supporting a linked mechanism between CLP1-promoted 3'UTR shortening and gene expression enhancement at Aire-sensitive genes.

## Clp1 promotes 3'UTR shortening and higher levels of Aire-upregulated transcripts in mTEChi

To test for the in vivo impact of Clp1 on 3'UTR length variation of the transcripts upregulated by Aire in mTEChi, we generated lentigenic *Clp1* knockdown mice. Three shRNAs targeting the murine *Clp1* with the highest knockdown efficiency (*Supplementary file 1*) were cloned as a multi-miR construct into a lentiviral vector, downstream of a doxycycline inducible promoter and upstream of the GFP as a marker of activity. Purified concentrated lentiviruses containing this construct and the lentiviral vector expressing the TetOn3G protein were used to microinfect fertilized oocytes, which were reimplanted into pseudopregnant females (*Figure 4A*). Of the 19 pups that we obtained with integration of both plasmids, two pups were expressing, after doxycycline treatment, the multi-miR in mTEChi and one pup was exhibiting a 60% reduction of *Clp1* mRNA levels in GFP+ (*Clp1* knockdown) versus GFP- (Ctr) mTEChi taken from the same mouse. These two cell populations, *Clp1* knockdown and no-knockdown Ctr mTEChi, were separated by GFP signal by FACS and processed for RNA-seq (*Figure 4—figure supplement 1*). We found a significant increase of d3'UTR ratios of Aire-sensitive genes following *Clp1* knockdown in mTEChi, as well as a less pronounced effect that does not reach significance for Aire-neutral genes regardless of whether they are TRA genes or not. This finding therefore supports that the effect of Clp1 on 3'UTR shortening preferentially affects Aire-sensitive genes in mTEChi (*Figure 4B* and *Figure 4—source data 1*). As in the HEK293 in vitro experiments, we observed that the d3'UTR ratios of the Aire-sensitive genes didn't reach the ratios of the Aire-neutral genes after *Clp1* knockdown, again due either to the incomplete 60% *Clp1*

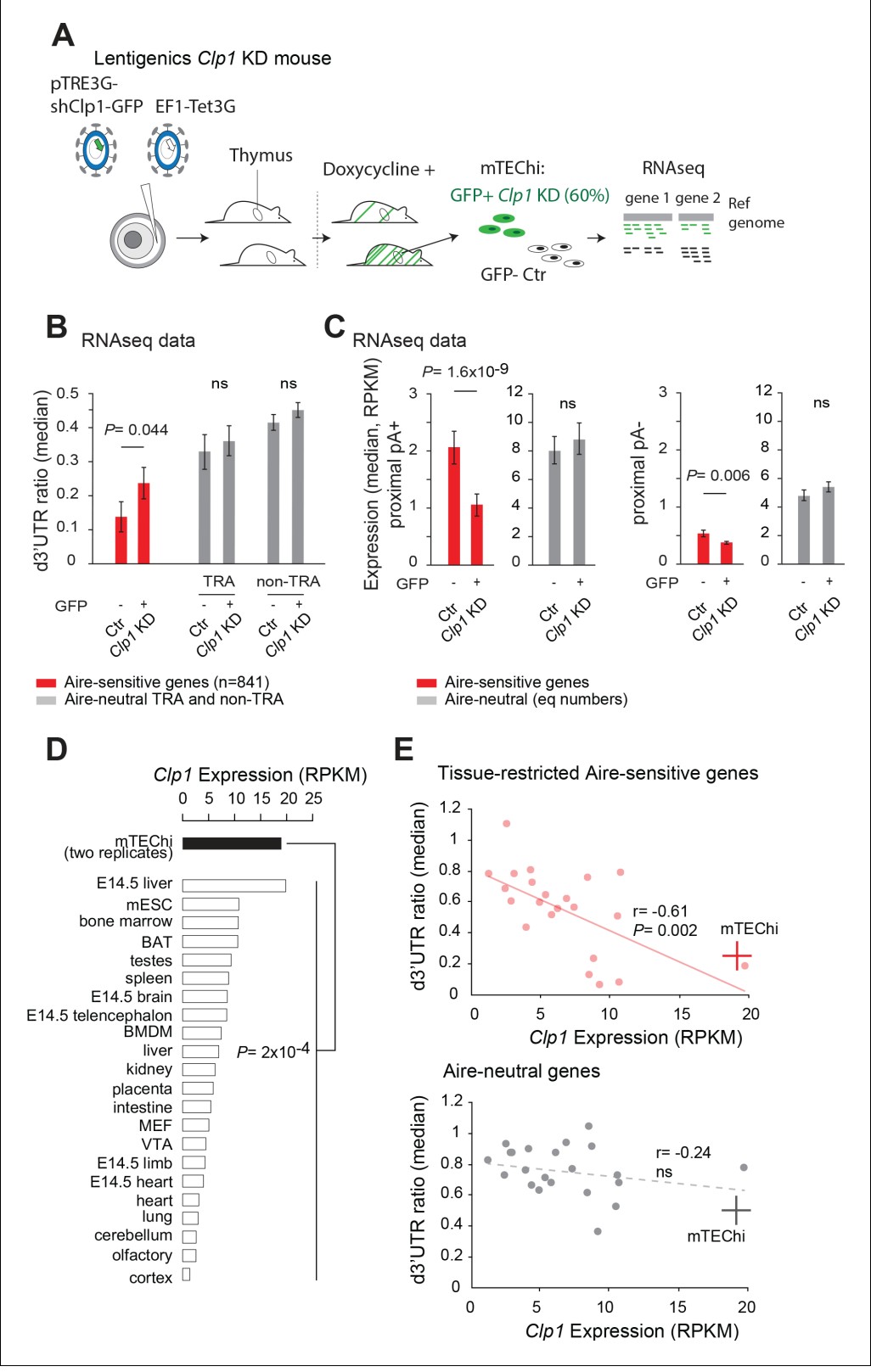

**Figure 4.** Clp1 controls the expression of Aire-upregulated genes with proximal pAs and their shortening in mTEChi. (**A**) Schematic of the lentigenic knockdown strategy. shRNAs against *Clp1* were transferred to a doxycycline-inducible expression system for microinfection of fertilized oocytes under the zona pellucid. The resulting pups were screened for integration of the constructs and, after treatment by doxycycline, for GFP

*Figure 4 continued on next page*

*Figure 4 continued*

expression in mTEChi. GFP+ and GFP- mTEChi were sorted and their transcripts profiled by RNA-seq. (B) Median of d3'UTR ratios of Aire-sensitive genes and all Aire-neutral genes characterized by a tissue-restricted expression (TRA genes) or not (non TRA genes) in GFP+ and GFP- mTEChi from a lentigenic mouse with GFP as a marker of *Clp1* knockdown activity; ns: non-significant sample-size adjusted P values. (C) Expression values of Aire-sensitive and neutral genes susceptible to differential pA usage (with potential proximal pAs: proximal pA+) (*Left*) and without potential proximal pAs (proximal pA-) in GFP+ and GFP- mTEChi (*Right*); equal numbers of neutral genes included. (D) *Clp1* expression from RNA-seq data of two replicate mTEChi and of 22 mouse tissues; median, log10 scale. BAT stands for brown adipocytes tissue, BMDM for bone marrow derived macrophage, MEF for mouse embryonic fibroblast, mESC for mouse embryonic stem cells and VTA for ventral tegmental area. Duplicate reads were discarded for datasets comparison. (E) Median of d3'UTR ratios of Aire-sensitive genes whose expression in the periphery is tissue-restricted and of Aire-neutral genes, relative to the levels of *Clp1* expression (log10 scale) in 22 mouse tissues. mTEChi are represented by a red and gray cross for the Aire-sensitive (*Top*) and neutral genes (*Bottom*), respectively; peripheral cells are represented by pink and gray circles. Significance is reached for the Aire-sensitive genes, p=0.002, Pearson correlation.

The online version of this article includes the following source data and figure supplement(s) for figure 4:

**Source data 1.** List of Aire-sensitive genes with proximal pAs in Ctr and *Clp1* KD mTEChi.
**Figure supplement 1.** Gating strategy used to purify GFP+ (*Clp1* knockdown) and GFP- (Ctr) mTEChi cells.
**Figure supplement 2.** Proportion of genes subject to Clp1-mediated 3'UTR shortening in Aire-sensitive and neutral genes in mTEChi.
**Figure supplement 3.** Aire-sensitive genes subject to Clp1-mediated 3'UTR shortening show higher expression in Ctr versus *Clp1* KD mTEChi.
**Figure supplement 4.** Comparison of gene expression in mTEChi and in mouse tissues.
**Figure supplement 5.** Clp1 expression in the thymus.
**Figure supplement 6.** Clp1 is not linked to nor controlled by Aire.

knockdown or to the existence of other non-Clp1-dependent differences. The preferential 3'UTR shortening effect of Clp1 at Aire-sensitive genes was further supported by the observation of an enhanced proportion of genes subject to Clp1-mediated 3'UTR shortening among the Aire-sensitive versus neutral genes in mTEChi (*Figure 4—figure supplement 2*).

We then compared the levels of expression of the Aire-sensitive genes with potential proximal pAs in Ctr versus *Clp1* knockdown mTEChi and found, in contrast to Aire-neutral genes, a significant reduction following *Clp1* knockdown (*Figure 4C*, *Left*). The link between Clp1-mediated 3'UTR shortening and increased expression at Aire-sensitive genes is strengthened by the observation that the Aire-sensitive genes that undergo a higher than 2-fold d3'UTR ratio decrease in Ctr versus *Clp1* knockdown lentigenic mTEChi, exhibit higher expression values (*Figure 4—figure supplement 3*). As observed in HEK293 cells, the effect of *Clp1* knockdown was dampened for the Aire-sensitive genes lacking potential proximal pAs (*Figure 4C*, *Right*). We also found that the expression of the genes without proximal pAs was globally reduced in comparison to the genes with proximal pAs, supporting a linked mechanism between proximal pA usage and increased gene expression in mTEChi. Altogether these results showed that Clp1 promotes 3'UTR shortening of the transcripts upregulated by Aire in mTEChi and strongly suggested that it enhances their level of expression through proximal pA usage.

A role for Clp1 in mTEChi was further supported by its higher expression in mTEChi in comparison to a wide variety of tissues for which we collected RNA-seq datasets (*Shen et al., 2012*; *van den Berghe et al., 2013*; *Warren et al., 2013*; *Figure 4D*). As a comparison, none of the other candidate factors, Hnrnpl, Ddx5, Ddx17 and Paprp1 that contributed to 3'UTR shortening in HEK293 cells, were over-represented in mTEChi (*Figure 4—figure supplement 4*). Importantly, we also validated higher expression of Clp1 at the protein level in mTEChi versus their precursor cells, mTEClo (*Gäbler et al., 2007*; *Hamazaki et al., 2007*), cells from the whole thymus, and also the predominant thymus CD45+ leukocyte fraction (*Figure 4—figure supplement 5*). In addition, a role for Clp1 independent of Aire's action on genes expression was supported by the observed similar levels of *Clp1* expression in WT and *Aire*-KO mTEChi (*Figure 4—figure supplement 6A*) and the lack of evidence for Aire and CLP1 interaction (*Figure 4—figure supplement 6B*). Finally, we found a significant correlation between higher *Clp1* expression and lower d3'UTR ratios across peripheral tissues for the Aire-sensitive TRA genes (*Figure 4E*, *Top*). In contrast, no such significant correlation was observed

for Aire-neutral genes (*Figure 4E*, *Bottom*). These findings indicate that the effect of Clp1 on 3'UTR shortening at Aire-sensitive genes is independent of Aire's action on gene expression, general across cell types, and conserved upon upregulation of gene expression by Aire in mTEChi.

## Clp1-driven 3'UTR shortening of Aire-upregulated transcripts is associated with higher stability in mTEChi

To determine whether the effect of Clp1 on 3'UTR shortening of Aire-upregulated transcripts was associated with higher levels of these transcripts in mTEChi through increased stability, we assessed the stability of all transcripts in these cells. We used Actinomycin D (ActD) to inhibit new transcription (*Sobell, 1985*) and assessed the differences in transcript profiles between treated and untreated mTEChi. We harvested the cells at several timepoints for expression profiling by RNA-seq. Each RNA-seq dataset was normalized to total read counts and we calculated the expression fold-change of each gene in treated versus control samples, therefore reflecting differences in transcript levels in the absence of ongoing transcription. We selected among the Aire-sensitive genes two subsets: (i) those that underwent a higher than 2-fold d3'UTR ratio decrease in Ctr versus *Clp1* knockdown lentigenic mTEChi and, (ii) genes with steady d3'UTR ratios (*Figure 4—source data 1*). Comparison of the two gene sets in ActD-treated versus control samples revealed that the Aire-sensitive genes subject to Clp1-driven 3'UTR shortening were initially comparable in their changes in transcript levels upon transcription inhibition to the changes for Aire-sensitive genes that showed no 3'UTR shortening but then showed increasing preservation of transcript levels at 3 hr and 6 or 12 hr, indicating a stabilization of this subset of 3'UTR-shortened transcripts (*Figure 5*; p=$2.1 \times 10^{-4}$, $1.3 \times 10^{-12}$, and $5.6 \times 10^{-10}$ at 3, 6 and 12 hr, respectively). Therefore, the Clp1-promoted 3'UTR shortening at Aire-sensitive genes in mTEChi is indeed associated with higher transcript stability.

Since short-3'UTR transcripts, in comparison to their longer counterparts, lack miRNA-binding sites specifically targeted by miRNAs known to destabilize transcripts and/or decrease translation, we assumed that the Aire-sensitive genes that underwent Clp1-promoted 3'UTR-shortening escape the post-transcriptional repression mediated by miRNAs in mTEChi. We assessed the level of miRNA-mediated post-transcriptional repression in mTEChi versus mTEClo precursor cells by assessing the differences in expression of sets of genes identified as potentially targeted by miRNAs, in mTEChi versus mTEClo. To this end, we performed a Gene Set Enrichment Analysis (GSEA) between the two mTEC populations for each set of genes potentially targeted by a specific miRNA or miRNA family (as defined by the TargetScan database, *Figure 5—source data 1*). We found 346 miRNA-specific target gene sets that had significantly reduced expression in mTEChi versus mTEClo (*Figure 5—figure supplement 1*, *Figure 5—source data 1*). In sharp contrast, we did not detect any of the miRNA-specific target gene sets that had reduced expression in mTEClo versus mTEChi. This suggests that miRNAs are an important contributor to transcript destabilization in mTEChi and that this effect is similar or less for the same miRNA target sets in mTEClo cells. To determine whether the Clp1-promoted 3'UTR shortening at Aire-sensitive genes can lead to escape from post-transcriptional repression mediated by the miRNAs associated with these 346 target-gene sets, we calculated, among the Aire-sensitive genes subject to Clp1-promoted shortening, the proportion of those that are potentially targeted by one of these miRNAs at a conserved site in the d3'UTR of their longer-transcript isoforms (*Figure 5—source data 1*). We found a large proportion of these cases (46%) had one or more such conserved miRNA target sites, providing a specific miRNA de-repression mechanism due to 3'UTR shortening for a large fraction of the Aire-sensitive genes. This analysis is only able to identify miRNAs that are highly conserved and that have disproportionate effect in mTEChi versus mTEClo, so it is expected to miss many of the cases of potential miRNA regulation that may be relieved by 3'UTR shortening in mTEChi.

## Discussion

Aire-sensitive genes upregulated by Aire in mTEChi have been shown to be controlled also by a number of Aire's allies or partners (*Abramson et al., 2010*; *Giraud et al., 2014*). Some of these factors have been reported to be part of the large 3' end processing complex (*de Vries et al., 2000*; *Shi et al., 2009*), notably Clp1 which belongs to the core 3' end processing sub-complex (*de Vries et al., 2000*; *Mandel et al., 2008*) and favors early cleavage and polyadenylation in some systems (*Holbein et al., 2011*; *Johnson et al., 2011*). In our present study, we demonstrated that Clp1

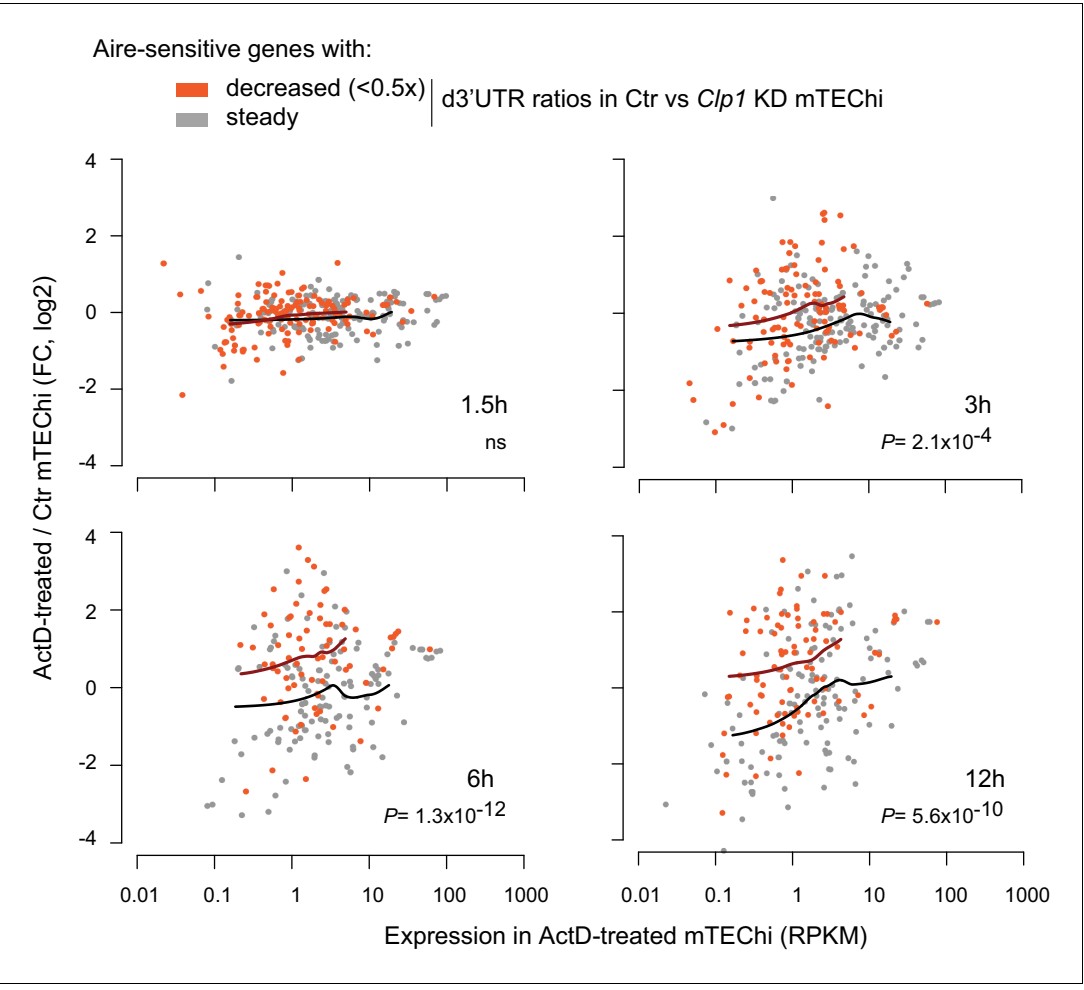

**Figure 5.** Clp1-driven 3'UTR shortening of the Aire-upregulated transcripts show higher stability in mTEChi. Relative expression of Aire-sensitive genes in ActD-treated (for indicated time durations) vs. control mTEChi depending on whether they undergo Clp1-mediated 3'UTR shortening as identified in *Clp1* lentigenics. Loess-fitted curves are shown in dark orange and black for the Aire-sensitive genes with decreased (<0.5 x) and steady d3'UTR ratios, respectively. P values are for comparison of expression ratios.

The online version of this article includes the following source data and figure supplement(s) for figure 5:

**Source data 1.** Potential miRNA-specific target genes with conserved miRNA sites in their 3'UTRs or d3'UTRs, from the TargetScan 6.2 database.

**Figure supplement 1.** miRNA-mediated post-transcriptional regulation in mTEChi versus mTEClo.

promotes 3'UTR shortening of the transcripts upregulated by Aire in mTEChi, and that these transcripts are associated with an enhanced stability, revealing a post-transcriptional mechanism whose escape leads to higher levels of expression of Aire-sensitive genes in mTEChi.

Comparison of RNA-seq expression profiles between *Clp1* knockdown and Ctr mTEChi isolated from *Clp1* lentigenic mice showed that Clp1 was able to promote 3'UTR shortening at Aire-sensitive genes, thereby contributing to the preference of short-3'UTR transcript isoforms of Aire-sensitive genes versus Aire-neutral genes in WT mTEChi. We found that this 3'UTR shortening was driven by Clp1 in mTEChi but also in Aire-non-expressing HEK293 cells, in which the level of Aire-sensitive gene expression is weak but still detectable by RNA-seq, thus showing that the effect of Clp1 on Aire-sensitive genes was not restricted to mTEChi nor dependent on Aire's action. Although Clp1 is a main contributor to this described process, additional factors might also be involved.

Clp1 is a ubiquitous protein showing higher expression in mTEChi than in mTEClo, CD45[+] thymic cells or thymic cells taken as a whole, but also showing higher gene expression in comparison with a

large range of peripheral tissue cells. Interestingly, we found that the level of *Clp1* expression was also significantly high in 14.5 embryonic liver cells, cells that have been reported to undergo sustained cellular activation leading to proliferation and maturation into hepatocytes (*Kung et al., 2010*). Similarities with the pattern of mTEC development (*Gray et al., 2007*; *Irla et al., 2008*) might point out cell activation as a potent inducer of Clp1 expression. Notably, we showed that higher levels of *Clp1* expression were correlated with higher proportions of short-3'UTR transcript isoforms of tissue-restricted Aire-sensitive genes across peripheral tissues, suggesting the existence of a Clp1-promoted 3'UTR shortening mechanism occurring in a variety of cell types and, among those cell types we surveyed, most pronounced in mTEChi.

The observation that the levels of *Clp1* expression and the distributions of the short-3'UTR transcript isoforms of Aire-sensitive genes were similar between *Aire*-KO and WT mTEChi, strongly suggests that the Clp1-driven 3'UTR shortening mechanism is already in place in mTEChi before Aire is activated. The independence of Aire's action on gene expression and the effect of Clp1 on 3'UTR shortening is also consistent with the lack of direct interaction found between Aire and Clp1. Although these effects of Aire and Clp1 appear decoupled, it is also apparent as noted that sensitivity to these Aire and Clp1 effects have a higher-rate of co-occurrence in the same set of genes. Although we could detect and characterize the effect of Clp1 on the Aire-sensitive genes in mTEChi using annotated pAs from the PolyA_DB 2 database which mainly contains pAs identified from comparisons across peripheral tissues, the precise proportion of Aire-sensitive genes that are subject to differential pA usage and Clp1-driven shortening in mTEChi remains to be precisely defined. Current methods to capture mTEChi-specific pAs using next generation sequencing methods require very large numbers of cells relative to the number of mTEChi (~30,000) that can be isolated per mouse but emerging methods for accurate pA identification with lower input requirements could make this feasible from such low material quantity (*Chen et al., 2017*).

Clp1 is a member of the core 3' end processing complex that we showed to be preferentially located at proximal pAs of Aire-sensitive genes in HEK293 cells, indicating that the effect of Clp1 on 3'UTR shortening could result from enhanced recruitment of the 3' end processing complex to proximal pAs. Similar modes of action, involving members of the core 3' end processing complex and resulting in transcripts with either shorter or longer 3'UTRs have been described for Clp1 in yeast (*Johnson et al., 2011*) and Cpsf6 in humans (*Gruber et al., 2012*; *Martin et al., 2012*), respectively. However, the basis for the preferential effect of Clp1 on the Aire-sensitive genes remains unknown, but note that it does appear to be conserved across cell types. One possibility is that the regulatory elements and associated basal transcriptional machinery at Aire-sensitive genes share conserved features that favor recruitment of Clp1.

3'UTRs have been described as potent sensors of the post-transcriptional repression mediated by miRNAs, resulting in mRNA destabilization and degradation (*Bartel, 2009*; *Jonas and Izaurralde, 2015*). In addition to miRNAs, RNA-binding proteins (RBPs) also contribute the regulation of transcript stability depending on the type of cis regulatory elements that they recognize on 3'UTRs, triggering either repression or activation signals (*Spies et al., 2013*). In mTEChi, we found increased stability of the Aire-sensitive transcripts that were subject to Clp1-promoted 3'UTR shortening. Thus, our findings supported an escape from the post-transcriptional repression of short-3'UTR transcripts in mTEChi, leading to enhanced stability and accumulation of these transcripts. Similar observations, resulting in increased transcript levels and higher protein translation, have notably been documented for genes subject to 3'UTR shortening whose long-3'UTR transcript isoforms were targeted by particular miRNAs or classes of RBPs (*Chen and Shyu, 1995*; *Guo et al., 2010*; *Masamha et al., 2014*; *Mayr and Bartel, 2009*; *Vlasova et al., 2008*).

The finding that the transcripts controlled by Aire and subjected to Clp1-promoted 3'UTR shortening in mTEChi escape the miRNA-mediated post-transcriptional repression that proves to be stronger in mTEChi than in mTEClo, is consistent with recent reports showing that maturation-dependent expression of miRNAs correlates with the expression of Aire in mTEChi (*Ucar et al., 2013*) and that Aire is able to control the expression of miRNAs in a murine mTEC line (*Macedo et al., 2015*) or in mTEChi in vivo through the comparison of WT and Aire null mutant thymi (*Ucar et al., 2013*). Interestingly, it has also been reported that, in contrast to Aire-independent TRA transcripts, some classical Aire-dependent TRAs exhibit profound refractoriness in reconstructed networking interactions with miRNAs (*Macedo et al., 2015*; *Oliveira et al., 2016*). An alteration in 3'UTRs of Aire-dependent TRA transcripts has been proposed to explain this

observation (*Passos et al., 2015*). Consistent with this proposal, our study provides evidence that Aire-sensitive genes undergo 3'UTR shortening resulting in the escape of the generated shortened transcripts from the miRNA-mediated post-transcriptional repression in mTEChi.

In addition to impacting transcript stability through the escape of the post-transcriptional repression, short 3'UTRs have also been shown to shift the surface localization of membrane proteins and the functional cell compartment of other types of proteins, in favor of the endoplasmic reticulum (ER) (*Berkovits and Mayr, 2015*). Thus, one interesting hypothesis for future study is that Clp1-driven 3'UTR shortening at Aire-sensitive genes in mTEChi might not only impact the expression of Aire-dependent self-antigens but also favor their routing to the ER, from where they will be processed and presented, potentially enhancing their presentation to self-reactive T lymphocytes and, subsequently, central tolerance and protection against autoimmune manifestations.

# Materials and methods

## Key resources table

| Reagent type (species) or resource | Designation | Source or reference | Identifiers | Additional information |
|---|---|---|---|---|
| Genetic reagent (*Mus-musculus*) | B6.129S2-Aire^tm1.1Doi/J | *Anderson et al., 2002* | RRID:IMSR_JAX:004743 | Provided by D. Mathis and C. Benoist (Harvard Medical School) |
| Cell line (*Homo-sapiens*) | HEK293 cells | *Martin et al., 2012* | | Cell line obtained from M. Zavolan lab |
| Cell line (*Mus-musculus*) | 1C6 mouse thymic epithelial cells | *Mizuochi et al., 1992* | | |
| Transfected construct (mouse) | pCMV-Aire-Flag plasmid | *Abramson et al., 2010* | | |
| Antibody | anti-mouse CD45-PerCPCy5.5 (Clone: 30-F11; Rat IgG2b, k) | Biolegend | Cat# 103131 RRID:AB_893344 | FACS (1:50) |
| Antibody | anti-mouse Ly51-PE (Clone: 6C3; Rat IgG2a, k) | Biolegend | Cat# 108307 RRID:AB_313364 | FACS (1:800) |
| Antibody | anti-mouse I-A/E-APC (Clone: M5/114.15.2; Rat IgG2b, k) | eBioscience | Cat# 17–5321 RRID:AB_469455 | FACS (1:1200) |
| Antibody | anti-Clp1 (Clone: EPR7181; Rabbit monoclonal) | GeneTex | Cat# GTX63930 | FACS (1:100) |
| Antibody | anti-human CLP1 (Goat polyclonal) | Santa Cruz | Cat# sc-243005 | IP and WB |
| Antibody | Flag-tag M2 (Mouse monoclonal) | Sigma | Cat# F1804 RRID:AB_262044 | IP and WB |
| Antibody | mouse CD45 microbeads | Miltenyi Biotec | Cat# 130-052-301 | |
| Recombinant DNA reagent | PLKO.1-Puro shRNAs (plasmids-lentiviruses) | Broadinstitute (RNAi platform) | | shRNA details: *Supplementary file 1* |
| Recombinant DNA reagent | EF1-Tet3G (plasmid-lentivirus) | Vectalys | | |
| Recombinant DNA reagent | pTRE3G-shClp1-GFP (plasmid-lentivirus) | Vectalys | | Clp1 shRNAs: *Supplementary file 1* |
| Sequence-based reagent | qPCR primers | this paper | | primer details: *Supplementary file 2* |

*Continued on next page*

*Continued*

| Reagent type (species) or resource | Designation | Source or reference | Identifiers | Additional information |
|---|---|---|---|---|
| Sequence-based reagent | oligo(dT)12–18 primers | ThermoFisher | Cat# 18418012 | |
| Commercial assay or kit | Trans-IT-293 transfection reagent | Mirus | Cat# MIR 2700 | |
| Commercial assay or kit | Universal Magnetic Co-IP Kit | Active Motif | Cat# 54002 | |
| Commercial assay or kit | Superscript II Reverse Transcriptase | ThermoFisher | Cat# 18064071 | |
| Chemical compound, drug | collagenase D | Roche | Cat# 11088866001 | (1 mg/mL final) |
| Chemical compound, drug | DNase I | Sigma | Cat# DN25 | (1 mg/mL final) |
| Chemical compound, drug | collagenase/dispase | Roche | Cat# 11097113001 | (2 mg/mL final) |
| Chemical compound, drug | actinomycin D | Sigma | Cat# A9415 | (1 µM) |
| Software, algorithm | Bowtie | Johns Hopkins University (http://bowtie-bio.sourceforge.net/index.shtml) | RRID:SCR_005476 | |
| Software, algorithm | Samtools | Samtools (http://samtools.sourceforge.net/) | RRID:SCR_002105 | |
| Software, algorithm | Bedtools | University of Utah (https://bedtools.readthedocs.io/en/latest/) | RRID:SCR_006646 | |
| Software, algorithm | DESeq | EMBL (http://bioconductor.org/packages/release/bioc/html/DESeq.html) | RRID:SCR_000154 | |
| Software, algorithm | RNAseq_d3UTR_ratio_density.R | this paper | | provided: *Figure 1—source data 3* |
| Software, algorithm | CEAS | CEAS (http://ceas.cbi.pku.edu.cn/index.html) | RRID:SCR_010946 | |
| Software, algorithm | aroma.affymetrix R package | UCSF (prev. UC Berkeley) (https://www.aroma-project.org) | RRID:SCR_010919 | details: *Figure 3—source data 1* |
| Software, algorithm | Microarray_d3UTR_hg19.R | this paper | | provided: *Figure 3—source data 2* |
| Software, algorithm | Microarray_d3UTR_significance.R | this paper | | provided: *Figure 3—source data 3* |
| Software, algorithm | GSEA | BroadInstitute (http://www.broadinstitute.org/gsea/) | RRID:SCR_003199 | |

## Mice

*Aire*-deficient mice on the C57BL/6 (B6) genetic background (*Anderson et al., 2002*) were kindly provided by D. Mathis and C. Benoist (Harvard Medical School, Boston, MA), and wild-type B6 mice were purchased from Charles River Laboratories. Mice were housed, bred and manipulated in specific-pathogen-free conditions at Cochin Institute according to the guidelines of the French Veterinary Department and under procedures approved by the Paris-Descartes Ethical Committee for

Animal Experimentation (decision CEEA34.MG.021.11 or APAFIS #3683 N° 2015062411489297 for lentigenic mouse generation).

## Cell culture

Cell lines used include HEK293 from the lab of Dr Zavolan and the murine thymic epithelial 1C6 cell line (*Mizuochi et al., 1992*). These cells were maintained in the lab, negative for mycoplasma and cultured in DMEM high glucose medium complemented with 10% FBS, L-glutamate, sodium pyruvate 1 mM, non-essential amino acids and pen/strep antibiotics. HEK293 cells display a normal HEK293 morphology and the identity of 1C6 cells was confirmed by RNA-seq with high expression of typical thymic epithelial markers.

## Isolation and analysis of medullary epithelial cells

Thymi of 4-wk-old mice were dissected, trimmed of fat and connective tissue, chopped into small pieces and agitated to release thymocytes. Digestion with collagenase D (1 mg/mL final) (Roche) and DNase I (1 mg/mL final) (Sigma) was performed for 30 min at 37°C. The remaining fragments were then treated with a collagenase/dispase mixture (2 mg/mL final) (Roche) and DNase I (2 mg/mL final) at 37°C until a single-cell suspension was obtained. Cells were passed through 70 μm mesh and resuspended in staining buffer (PBS containing 1% FBS and 5 mM EDTA). For isolation of medullary epithelial cells, an additional step of thymocyte depletion was performed using magnetic CD45 MicroBeads (Miltenyi Biotec). The resuspended cells were incubated for 20 min at 4°C with the fluorophore-labeled antibodies CD45-PerCPCy5.5 (1:50) (Biolegend), Ly51-PE (1:800) (Biolegend), and I-A/E-APC (1:1,200) (eBioscience). Sorting of mTEChi/lo (CD45$^-$Ly51$^-$I-A/E$^{high/low}$) or, for lentigenic mice, of mTEChi +/- for GFP expression, was performed on a FACSAria III instrument (BD Bioscience). For Clp1 staining, cells labeled for membrane antigens were fixed in (3.7%) formaldehyde for 15 min, permeabilized in (0.5%) saponin for 15 min, and incubated with an antibody to Clp1 (1:100) (clone: EPR7181, GeneTex, GTX63930) and an Alexa Fluor 488-conjugated goat polyclonal antibody to rabbit (1:200) (TermoFisher, A11008). Cells were analyzed on an Accuri C6 instrument (BD Bioscience). All compensations were performed on single-color labeling of stromal cells and data analysis was done using the BD Accuri C6 Analysis software.

## Actinomycin D treatment

mTEChi were isolated and sorted (~4 x 10$^5$) from pooled thymi of B6 mice as described above, then treated with actinomycin D (1 μM) in MEM medium for 3, 6 and 12 hours at 37°C. The cells were then harvested and total RNA was isolated by TRIzol extraction (ThermoFisher).

## *Aire*-transient transfections

HEK293 cells were seeded at a density of 600,000 per well in 6-well plates or at 3.5*10$^6$ per 10-cm$^2$ culture dish. The next day, and depending on the dish format, HEK293 cells were transfected with either 2.5 or 8 μg of the pCMV-Aire-Flag plasmid (or control plasmid) using 7.5 or 32 μl of the TransIT-293 transfection reagent (Mirus). After 48 hr, cells cultured in 6-well plates were subjected to total RNA extraction for RNA-seq experiments, whereas those in the culture dish were subjected to protein extraction for coimmunoprecipitation.

## Coimmunoprecipitation

Extraction of nuclear proteins and coimmunopreciptation were performed using the Universal Magnetic Co-IP Kit (Active Motif). Briefly, *Aire*-transfected HEK293 cells were lysed with hypotonic buffer and incubated on ice for 15 min. Cell lysates were centrifuged for 30 s at 14,000 x g, then the nuclei pellets were digested with an enzymatic shearing cocktail for 10 min at 37°C. After centrifugation of the nuclear lysates, the supernatants containing the nuclear proteins were first incubated with specific antibodies for 4 hr, then with Protein-G magnetic beads for 1 hr at 4°C with rotation. After four washes, bound proteins were eluted in laemmli/DTT buffer, separated by SDS/PAGE for 40 min at 200 V, transferred to PVDF membranes using the TurboTransfer System for 7 min at 25 V (BioRad) and blocked for 1 hr with TBS, 0.05% Tween, 3% milk. The western blot detection was done after incubation with primary (2 hr) and secondary antibodies (1 hr). Detection was performed by enhanced chemiluminiscence (ECL). Antibodies used for immunoprecipitation or revelation were:

CLP1 (sc-243005, Santa Cruz), Flag-tag M2 (F1804, Sigma), goat IgG control (sc-2028, Santa Cruz), mouse IgG1 control (ab18443, Abcam), and horseradish peroxidase-conjugated anti-mouse IgG light chain specific (115-035-174, Jackson ImmunoResearch).

## shRNA-mediated knockdown

Specific knockdown of *CLP1*, *HNRNPL*, *DDX5*, *DDX17*, *PARP1*, *PRKDC*, *SUPT16H*, *PABPC1*, *CPSF6* and the control LacZ gene in HEK293 cells, or of *Clp1* in the 1C6 mouse thymic epithelial cell line was performed by infection with lentivirus-expressing shRNAs. shRNAs were cloned into the lentivirus vector pLKO with an expression driven by the ubiquitously active U6 promoter, and were provided by the RNAi Consortium of the Broad Institute, as lentiviral particles at ~$10^7$ VP/mL. For each candidate, we tested an average of 5 specific shRNAs among those with the highest knockdown efficiency as measured by the RNAi platform of the Broad Institute.

HEK293 or 1C6 cells were seeded at a density of 250,000 or 650,000 per well in 6-well plates. The next day, cells were supplemented with 8 mg/ml polybrene and infected with 20 µL of shRNA-bearing lentiviruses. Each shRNA was tested in duplicate. The day after, the medium was changed to a fresh one containing 2 µg/ml puromycin. Cells were maintained in selective medium during 6 days and harvested for RNA extraction using TRIzol (ThermoFischer). Knockdown efficiency was analyzed by real-time PCR – carried out in technical triplicate – in comparing the level of expression of each candidate in the knockdown vs. control samples using the human or murine GAPDH gene for normalization with primers listed in *Supplementary file 2*. First-strand cDNA was synthesized using SuperScript II Reverse Transcriptase (ThermoFischer) and oligo(dT)12–18 primers. cDNA was used for subsequent PCR amplification using the 7300 Real-Time PCR system (Applied Biosystems) and SYBR Green Select Master Mix (ThermoFisher). Knockdown efficiency of each specific shRNA was summarized in *Supplementary file 1*. We used for subsequent analyses the extracted RNA corresponding to the three shRNAs yielding the highest reduction of their target mRNA (>60%).

## Lentigenic mouse generation

Three shRNAs against *Clp1* were cloned into a cluster of micro RNAs construct, the two most efficient shRNAs in two copies and the third in a single copy (*Supplementary file 1*). This construct was transferred to a lentiviral backbone, downstream of the doxycycline inducible promoter TRE3G and upstream of the ZsGreen protein. A second construct expressing the TetOn3G transactivator under the control of the EF1 promoter was generated. A single ultra-high purified and concentrated lentivector (2.2 × $10^9$ TU/mL) containing both constructs was generated and purified by both Tangential Flow Filtration and Chromatography. Cloning and lentiviral production were performed by Vectalys (www.vectalys.com). Fertilized oocytes (B6) were microinjected under the zona pellucida with the lentivirus suspension as described (*Giraud et al., 2014*). A pool of 33 transduced oocytes were reimplanted into five pseudopregnant females. Newborns were selected for integration of both constructs by PCR with primers matching the ZsGreen or TetOn3G sequences (*Supplementary file 2*). At 3 weeks of age, mice were treated with doxycycline food pellets (2 g/kg) for two weeks and then sacrificed for mTEChi isolation. Mice expressing GFP in >10% of mTEChi were selected for GFP+ and GFP- mTEChi RNA-seq.

## RNA-seq and d3'UTR ratios

Total RNA was isolated by TRIzol extraction (ThermFisher). Only high-quality RNA (with RIN values around or over 8) was used to generate polyA-selected transcriptome libraries with the TruSeq RNA Library Prep Kit v2, following the manufacturer's protocol (Illumina). Sequencing was performed using the Illumina HiSeq 1000 machine and was paired-end (2 × 100 bp) for mTEChi and *Aire*-KO mTEChi isolated from pooled thymi (deposited in the GEO database as GSE140683), and for mTEChi and mTEClo isolated from the same mice (deposited in the GEO GSE140815). Sequencing was single-end (50 bp) for transfected and knockdown HEK293 cells (deposited in the GEO database as GSE140738 and GSE140993), actinomycin D-treated mTEChi (as GSE140815), and mTEChi isolated from *Clp1* lentigenic mice (as GSE140878). RNA libraries from thymic cells isolated from lentigenic mice or actinomycin D-treated mTEChi were constructed with the Smarter Ultra Low Input RNA kit (Clontech) combined to the Nextera library preparation kit (Illumina). Paired-end (100 bp) datasets were homogenized to single-end (50 bp) data by read-trimming and concatenation. Lower quality

reads tagged by the Illumina's CASAVA 1.8 pipeline were filtered out and mapped to the mouse or human reference genome (mm9 or hg19) using the Bowtie aligner (*Langmead et al., 2009*). For the multi-tissue comparison analysis, duplicate reads were removed with the Samtools rmdup function (*1000 Genome Project Data Processing Subgroup et al., 2009*). For read counting, we used the intersectBed and coverageBed programs of the BEDtool distribution (*Quinlan and Hall, 2010*) with the -f one option. It enabled the count of reads strictly contained in each exon of the Refseq genes whose annotation GTF file was obtained from the UCSC Table Browser, in choosing mm9 mouse (or hg19 human) genome/Genes and Gene Predictions/RefSeq Genes/refGene. GTF files for mice and humans are provided in *Figure 1—source data 2*. Differential fold-change expression between two datasets was computed using DESeq (*Anders and Huber, 2010*) and gene expression was quantified in each sample as reads per kilobase per million mapped reads (RPKM).

For d3'UTR ratio calculation, we counted the reads as above but with a GTF file that we generated as follows: First, from the UCSC Table Browser, we chose mm9 mouse (or hg19 human) genome/Genes and Gene Predictions/RefSeq Genes/refGene, and we obtained one file with coding exon features only and another file with 5' and 3'UTR features. We combined these two files into a single one and split the 3'UTR annotations into distal 3'UTR (d3'UTR) and proximal 3'UTR annotations at the sites of alternative pAs that we identified in parsing the PolyA_DB 2 database (http://exon.umdnj.edu/polya_db/v2/). A proximal pA was validated when its genomic location from the PolyA_DB 2 database differs from 20 bp at least to the genomic location of the UCSC annotated 3'UTR distal boundary or distal pA. In case of multiple proximal pAs, the most proximal one was considered. Then, we fused into a single feature the proximal 3'UTR annotations with the last coding exons (exon CDS) and formatted the file to fit a GTF format. d3'UTR annotation and GTF files for mice and humans are provided in *Figure 1—source data 1*. For each Refseq gene corresponding to an independent NM_Refseq ID, we divided the RPKM expression of the d3'UTR region by the expression of the upstream region of the last 3' exon to generate the d3'UTR ratio used as a measure of pA usage. Note that using two contiguous regions in the last 3' exons of the genes for d3'UTR ratio calculation, reduces the risk of a potential bias that could arise from a non-uniform RNA-seq coverage between the 3' ends and their upstream regions. The R-script used for d3'UTR ratio calculation in WT and *Aire*-KO mTEChi is provided in *Figure 1—source data 3*.

## Multi-tissue comparison analysis

First, an RNA-seq database of mouse tissues was assembled in collecting 22 RNA-seq datasets generated from polyA-selected RNA and Illumina sequencing. The raw read sequences were obtained from the GEO database: GSE36026 for bone marrow, bone marrow derived macrophage (BMDM), brown adipocytes tissue (BAT), cerebellum, cortex, heart, intestine, kidney, liver, lung, olfactory, placenta, spleen, testes, mouse embryonic fibroblast (MEF), mouse embryonic stem cells (mESC), E14.5 brain, E14.5 heart, E14.5 limb and E14.5 liver; GSM871703 for E14.5 telencephalon; GSM879225 for ventral tegmental area VTA. Reads of the VTA dataset, over 50 bp in length, were trimmed for homogeneous comparison with our RNA-seq data. We processed each collected dataset for gene expression profiling and d3'UTR ratio computing as above. To avoid center-to-center biases, we removed from the sequence assemblies the duplicate reads that could arise from PCR amplification errors during library construction. For multi-sample comparison analysis, our mTEChi datasets were also subjected to removal of duplicate reads.

Next, the tissue-specificity (one tissue of restricted expression) or selectivity (two-to-four tissues of restricted expression) of the Aire-sensitive genes was characterized by using the specificity measurement (SPM) and the contribution measurement (CTM) methods (*Pan et al., 2013*). For each gene, the SPM and the CTM values were dependent on its level of expression in each tissue. If no read was detected in a tissue, the latter was excluded from the comparison. For tissues of similar type, only the one showing the highest level of gene expression was included in the comparison. Cerebellum, cortex, E14.5 brain, E14.5 telencephalon and VTA referred to a group, as well as did E14.5 heart and heart, or E14.5 liver and liver. If the SPM value of a gene for a tissue is >0.9, then the gene is considered tissue-specific for this particular tissue. Otherwise, if the SPM values of a gene for two to four tissues were >0.3 and its CTM value for the corresponding tissues was >0.9, then the gene was considered tissue-selective for these tissues. If these conditions were not met, the gene was left unassigned.

Finally, for the analysis of 3'UTR length variation of Aire-sensitive PTA genes between mTEChi and their tissues of expression, peripheral d3'UTR ratios of tissue-specific genes were selected in their unique identified tissues of expression. For tissue-selective genes, the d3'UTR ratios for peripheral tissues were selected randomly among their tagged tissues of expression.

## CLIP-seq analysis

The location of the 3' end processing complex on the transcripts of Aire target genes was tracked by measuring the density of reads that map to these transcripts in PAR-CLIP sequencing data of RNAs bound to the endogenous CSTF2 from the GEO database (GSM917676). We processed these assemblies of RNA-mapped reads (as wig files) using the sitepro program (CEAS distribution) (*Shin et al., 2009*) to infer the read density at the vicinity of proximal and distal pAs of transcripts of Aire target genes in HEK293 cells. The genomic location of these proximal and distal pAs extracted from the PolyA_DB 2 database (*Figure 2—source data 1*) was loaded into sitepro as bed files of Aire-sensitive and neutral genes identified from RNA-seq differential expression of *Aire* versus control-transfected HEK293 cells.

## Microarray gene expression profiling

Total RNA was prepared from harvested HEK293 cells knockdown for *CLP1*, *HNRNPL*, *DDX5*, *DDX17*, *PARP1*, *PRKDC*, *SUPT16H*, *PABPC1*, *CPSF6* and the control LacZ gene using TRIzol (ThermoFisher). Single-stranded DNA in the sense orientation was synthesized from total RNA with random priming using the GeneChip WT Amplification kit (Affymetrix). The DNA was subsequently purified, fragmented, and terminally labeled using the GeneChip WT Terminal Labeling kit (Affymetrix) incorporating biotinylated ribonucleotides into the DNA. The labeled DNA was then hybridized to Human Gene ST1.0 microarrays (Affymetrix), washed, stained, and scanned. Raw probe-level data (.CEL files deposited in the GEO database as GSE141118) was normalized by the robust multiarray average (RMA) algorithm and summarized using the R-package aroma.affymetrix (www.aroma-project.org/) to obtain probeset expression values corresponding to the genes on the array, as described in (*Figure 3—source data 1*).

## Individual probe-level microarray analyses

For each HEK293 sample of a microarray comparison, an expression file of individual probes of each gene on the array was generated using aroma.affymetrix, as described in (*Figure 3—source data 1*). Probes with expression values over the background in each sample, had their hg19 genomic location retrieved from the Affymetrix individual probe dataset (NetAffx Web site), and were mapped to the d3'UTR annotation file generated as described in the 'RNA-seq and d3'UTR ratios' section. The d3'UTRs ratios were then computed for the genes having at least two individual probes in their d3'UTR regions by dividing the specific d3'UTR expression by the whole-transcript expression. We performed the individual probe d3'UTR mapping and d3'UTR ratio calculation using an adaptation of our R-implementation of PLATA (*Giraud et al., 2012*), as described in *Figure 3—source data 2*. We then tested between two HEK293 samples the proportion of genes with a significant increase or decrease of d3'UTR ratios to the proportion of those whose variation is not significant (*Figure 3—source data 3*).

## miRNA potential target gene identification

We used the TargetScan 6.2 database (http://www.targetscan.org/vert_61/) to generate a list of miRNAs or miRNA families and their potential target genes harboring miRNA conserved sites in 3'UTRs (*Friedman et al., 2009*). We further selected from these potential target genes those for which the miRNA conserved sites are specifically located within the cleavable d3'UTRs. The miRNAs and their potential target genes with miRNA conserved sites within their 3'UTRs or d3'UTRs are provided in *Figure 5—source data 1*.

## Gene set enrichment analysis (GSEA)

GSEA was first used to test the effect of candidate gene knockdown on the expression of Aire-sensitive genes. The overlap between the transcripts impacted by Aire in HEK293 cells and those impacted by the knockdown of *CLP1*, *HNRNPL*, *DDX5*, *DDX17*, *PARP1* was tested by the GSEA

software (*Subramanian et al., 2005*) (Broad Institute). The Aire-sensitive genes were identified from RNA-seq data of control and *Aire*-transfected HEK293 cells. For this analysis, the top 30% of genes the most sensitive to Aire were considered.

GSEA was also used to test the potential involvement of miRNAs or miRNA families in the regulation of sets of genes in mTEChi versus mTEClo. To this end, we performed RNA-seq profiling of mTEChi and mTEClo that were isolated from the same pool of 4 thymi of B6 mice. We ran GSEA on the DESeq-normalized read counts of the mTEChi versus mTEClo comparison using the list of miRNAs and of their potential target genes harboring miRNA conserved sites in 3'UTRs as generated through parsing the TargetScan 6.2 database (above). A 'classic' scoring scheme was used for the GSEA analysis. Gene sets of 15 to 5000 genes were considered for the analysis. Null read count values were removed. To determine the genes that contribute the most to the observed miRNA-specific gene set reduction in mTEChi, we performed a leading edge analysis in including all miRNA-specific gene sets with FDR q-values <0.05 (*Figure 5—source data 1*). We found that these leading edge genes were potentially targeted by a number of different miRNAs, whose median was 4. We selected for representation the leading edge genes targeted by more than four different miRNAs (>=5).

## Statistical analysis

Determination of the statistical significance differences between two experimental groups was done by the non-parametric Wilcoxon test, unless specified. Since P-values are sensitive to sample size and tend to decrease as n increases, for each comparison, we matched the size of the control sample (when larger than the test sample) to the one of the test sample. For instance, for the Aire-sensitive versus neutral gene comparisons, we selected a number of Aire-neutral genes with the closest-to-1 expression foldchange, that matches the size of the tested Aire-sensitive sample.

## Acknowledgements

We thank Dr Sheena Pinto for her useful comments and suggestions, that have helped improve this paper. We thank Drs. D Mathis and C Benoist (Harvard Medical School) for *Aire*-KO (B6) mice. We thank the members of the 'Homologous recombination' and the 'Genomic'IC' core facilities (Cochin Institute) for lentigenic generation and RNA-seq data production. This work was supported by the Agence Nationale de la Recherche (ANR) 2011-CHEX-001-R12004KK (to MG), the European Commission CIG grant PCIG9-GA-2011–294212 (to MG) and by the 'Investissements d'Avenir' program managed by the ANR to the France Génomique national infrastructure ANR-10-INBS-09 (FC) and the French Institute of Bioinformatics ANR-11-INBS-0013 (CB). CG and Y-CL were supported by fellowships from the Fondation pour la Recherche Médicale FDT20150532551 and ING20121226316, respectively. CG, NJ and MG designed the study and wrote the manuscript; CG and NJ performed most of the experimental work; FP performed bioinformatics analyses and experiments, notably co-immunoprecipitations; FP, Y-CL and MG performed bioinformatics analyses; CB provided us with computing resources on the IFB national service infrastructure in bioinformatics and help with script optimization, OU and NF with RNA-seq and microarray datasets, and DER with lentivirus materials and editing of the manuscript.

## Additional information

### Funding

| Funder | Grant reference number | Author |
| --- | --- | --- |
| Agence Nationale de la Recherche | Research grant 2011-CHEX-001-R12004KK | Matthieu Giraud |
| European Commission | Career Integration Grant PCIG9-GA-2011-294212 | Matthieu Giraud |
| Agence Nationale de la Recherche | Investissements d'Avenir ANR-10-INBS-09 | Fanny Coulpier |
| Agence Nationale de la Recherche | Investissements d'Avenir ANR-11-INBS-0013 | Christophe Blanchet |

| Fondation pour la Recherche Médicale | Graduate Student Fellowship FDT20150532551 | Clotilde Guyon |
| --- | --- | --- |
| Fondation pour la Recherche Médicale | Bioinformatics engineer grant ING20121226316 | Yen-Chin Li |

The funders had no role in study design, data collection and interpretation, or the decision to submit the work for publication.

### Author contributions

Clotilde Guyon, Nada Jmari, Conceptualization, Investigation, Methodology; Francine Padonou, Formal analysis, Investigation; Yen-Chin Li, Christophe Blanchet, Formal analysis; Olga Ucar, Noriyuki Fujikado, Fanny Coulpier, Investigation; David E Root, Validation, Methodology; Matthieu Giraud, Conceptualization, Resources, Formal analysis, Supervision, Funding acquisition, Validation, Investigation, Methodology, Project administration

### Author ORCIDs

Matthieu Giraud https://orcid.org/0000-0002-1208-9677

### Ethics

Animal experimentation: Mice were housed, bred and manipulated in specific-pathogen-free conditions at Cochin Institute according to the guidelines of the French Veterinary Department and under procedures approved by the Paris-Descartes Ethical Committee for Animal Experimentation (decision CEEA34.MG.021.11 or APAFIS #3683 No 2015062411489297 for lentigenic mouse generation).

### Decision letter and Author response

Decision letter https://doi.org/10.7554/eLife.52985.sa1
Author response https://doi.org/10.7554/eLife.52985.sa2

# Additional files

### Supplementary files

• Supplementary file 1. List of shRNAs.

• Supplementary file 2. List of primers.

• Transparent reporting form

### Data availability

All RNAseq and microarray data are deposited in the NCBI Gene Expression Omnibus database (GEO).

The following datasets were generated:

| Author(s) | Year | Dataset title | Dataset URL | Database and Identifier |
| --- | --- | --- | --- | --- |
| Giraud M, Guyon C | 2020 | WT and Aire-KO mouse MEChi RNAseq profiling | https://www.ncbi.nlm.nih.gov/geo/query/acc.cgi?acc=GSE140683 | NCBI Gene Expression Omnibus, GSE140683 |
| Giraud M, Jmari N, Padonou F | 2020 | RNAseq profiling of Aire and Ctr-transfected HEK293 cells | https://www.ncbi.nlm.nih.gov/geo/query/acc.cgi?acc=GSE140738 | NCBI Gene Expression Omnibus, GSE140738 |
| Giraud M, Guyon C | 2020 | Stability of Aire-upregulated and Aire-neutral transcripts in medullary thymic epithelial cells | https://www.ncbi.nlm.nih.gov/geo/query/acc.cgi?acc=GSE140815 | NCBI Gene Expression Omnibus, GSE140815 |
| Giraud M, Guyon | 2020 | Clp1 knockdown lentigenic mouse | https://www.ncbi.nlm. | NCBI Gene |

| | | | | nih.gov/geo/query/acc.cgi?acc=GSE140878 | Expression Omnibus, GSE140878 |
| C, Jmari N | | | generation | | |
| Giraud M, Jmari N | 2020 | RNAseq profiling of Ctr and CLP1 knockdown HEK293 cells | | https://www.ncbi.nlm.nih.gov/geo/query/acc.cgi?acc=GSE140993 | NCBI Gene Expression Omnibus, GSE140993 |
| Giraud M, Jmari N | 2020 | RNAseq profiling of Ctr and CLP1 knockdown HEK293 cells | | https://www.ncbi.nlm.nih.gov/geo/query/acc.cgi?acc=GSE141118 | NCBI Gene Expression Omnibus, GSE141118 |

The following previously published datasets were used:

| Author(s) | Year | Dataset title | Dataset URL | Database and Identifier |
|---|---|---|---|---|
| Gruber AR | 2012 | PAR-CLIP CstF-64 | https://www.ncbi.nlm.nih.gov/geo/query/acc.cgi?acc=GSM917676 | NCBI Gene Expression Omnibus, GSM917676 |
| Abramson J, Giraud M | 2016 | Aire-KO MEChi RNAseq profiling | https://www.ncbi.nlm.nih.gov/geo/query/acc.cgi?acc=GSE87133 | NCBI Gene Expression Omnibus, GSE87133 |
| Abramson J, Giraud M | 2015 | Sirt1 is essential for Aire-mediated induction of central immunological tolerance | https://www.ncbi.nlm.nih.gov/geo/query/acc.cgi?acc=GSE68190 | NCBI Gene Expression Omnibus, GSE68190 |

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
