## [Decision Letter]

**Acceptance summary:**

In this manuscript, Giraud and colleagues address the detailed mechanism via which medullary thymic epithelial cells (mTECs) express a wide range of otherwise tissue restricted genes. This process is required to ensure that only a self-tolerant T cell repertoire leaves the thymus to populate the peripheral immune system, and is largely regulated by the Autoimmune Regulator, AIRE. The authors find that AIRE-sensitive genes preferentially exhibit short-3'UTR transcript isoforms in mTECs. They then demonstrate that CLP1, part of the large multi-subunit 3' end processing complex, promotes 3'UTR shortening associated with higher transcript stability and expression of AIRE-sensitive genes, revealing a previously uncovered post-transcriptional level of control of AIRE activated expression in mTECs.

**Decision letter after peer review:**

Thank you for submitting your article "Aire-dependent genes undergo Clp1-mediated 3'UTR shortening associated with higher transcript stability in the thymus" for consideration by *eLife*. Your article has been reviewed by three peer reviewers, and the evaluation has been overseen by a Reviewing Editor and Satyajit Rath as the Senior Editor. The reviewers have opted to remain anonymous.

The reviewers have discussed the reviews with one another and the Reviewing Editor has drafted this decision to help you prepare a revised submission.

Summary:

Overall, the reviewers found this manuscript interesting and felt that it has potential to be a significant contribution to the field. However, they identified a number of substantive points that must be satisfactorily addressed before the manuscript can be considered for publication in *eLife*. In particular, the reviewers felt that the current evidence presented is not yet strong enough to support the author's proposed mechanism. Concerns were raised that the evidence is quite descriptive that these 3'UTR shortening events are associated with transcript stability. Additionally, concerns were raised about bioinformatics methods throughout the manuscript. Finally, some parts of the manuscript need rewriting to make the discussion of existing literature more comprehensive.

Essential revisions:

Taken together, the major points raised by the reviewers can be broken into three areas – required further biological insight; clarification/improvement of bioinformatics analyses; and textual changes, as follows:

Biological insight: A number of points were raised relating to the need for further evidence to support the authors' main conclusions.

Two reviewers raised issues relating to whether or not Clp1 3'UTR shortening is a mechanism related to Aire's function or may likewise affect Aire independent genes (See title of the paper). This important point should be fully addressed in a revised manuscript, including via answers to the specific points raised below:

– To support the claim that 3'UTR shortening is due to Clp1, the authors are asked to address in more detail whether the Aire-sensitive genes influenced by Aire-KO and Clp1-KD are mostly shared or with the similar pattern around pA sites. If not, why?

– The data presented all point towards use of proximal pA sites through the action of Clp1 being an independent layer of control of TRA expression (in the sense of being 'upstream' of Aire's function rather than being a consequence of Aire's action). mTECs are thought to express 'many' TRAs in an Aire independent manner (or do the authors feel otherwise?). Are these also affected by knockdown of Clp1? In sum, the question arises whether 'Aire dependent genes undergo Clp1-mediated 3'UTR shortening….' (as the title insinuates) or whether 'Weakly expressed genes undergo Clp1-mediated 3'UTR shortening…'

Two reviewers also asked for further data related to the role of Clp1 in mTECs. Again, these points should be addressed in a revised manuscript.

– Please provide additional data on how Clp1-driven 3'UTR shortening can lead to high transcript stability. Are any specific microRNAs or RNA-binding proteins involved during this process?

– Unfortunately, the authors only address the effect of Clp1 knockdown in the absence of Aire in HEK293 cells or in the presence of Aire in mTECs. It would be extremely interesting to relate this to the effect-size of Aire in HEK293 cells and in mTECs (e.g. how would Figure 3D, right, look like for Aire-expressing HEK293 cells in which Clp1 has been knocked-down). Alternatively (but certainly asking too much), it would have been very informative to see how Figure 4B, right, would look like for *Aire*-KO mTECs).

Bioinformatics: The reviewers raised a number of concerns and questions regarding bioinformatics methods and analyses, which should be fully addressed in a revised manuscript.

1) Please provide more detail on how the d3'UTR ratio, used to measure 3'UTR usage, was calculated. Did the method used consider non-uniform read bias? Whether d3'UTR ratio is based on transcript or genes? Also, please explain why d3'UTR ratio has a maximum value of 2 based on Figure 1C, E, Figure 2A etc?

2) Equal number Aire-neutral datasets were used as the control throughout the manuscript. The authors are asked to address why this small arbitrarily selected dataset can represent the global pattern of Aire-neutral genes. They should describe how the Aire-neutral genes were selected, and whether these are equally low in their basal expression levels (see below).

3) Aire-sensitive genes are generally expressed at very low levels as compared to most other genes (= Aire-insensitive/neutral genes) (see also Figure 1B). Please comment on whether or not in the case of 'low abundance' mRNAs, there might be a general technical bias towards detection of more 5' reads even among reads that are all upstream of the proximal pA site (which would result in an artificially low d3'UTR ratio). In this context, are the selected Aire neutral genes on average expressed at a similarly low level (see for instance Figure 4B, right: here, the median expression (RPKM) of Aire-neutral genes is substantially higher than that of Aire-sensitive genes; see also Figure 3D)?

4) Related to Figure 3A, the authors are asked to explain why some known APA factors did not change d3'UTR ratio.

5) For the multi-tissues analysis from various sources, did the authors consider batch effect and other technical bias? Specifically, related to Figure 1G, an important question is whether a gene that shows a short 3'UTR preference in mTECs also does so in the peripheral tissue. Is that what Figure 1G is supposed to address? It is very difficult to understand. Would it be feasible to plot for all the individual 574 Aire-sensitive TRAs from Figure 1D their d3'UTR ratio in mTEChi vs. their d3'UTR ratio in the respective peripheral tissue to ask in how far the distribution of datapoints deviates from a diagonal? Possibly, this might be a more intuitive way of showing whether or not the usage of a proximal 3' UTR differs between mTEChi and a respective peripheral tissue.

6) Please ensure that all of the bioinformatics methods and tools used are appropriately referenced, as this is currently not the case.

Textual changes:

Concerns were raised that the existing literature is only incompletely discussed, particularly in the Introduction. Please include further discussion of the current literature around post-transcriptional control mechanisms in mTECs involving changes in the mRNA 3´UTRs, and fully cite relevant previous references.

Also, subsection “Aire-sensitive genes show a preference for short-3’UTR transcript isoforms in mTEChi and in some peripheral tissues”, last paragraph, first sentence: This somewhat contrived sentence may deserve attention.

---

## [Author Response]

Essential revisions:Taken together, the major points raised by the reviewers can be broken into three areas – required further biological insight; clarification/improvement of bioinformatics analyses; and textual changes, as follows:Biological insight: A number of points were raised relating to the need for further evidence to support the authors' main conclusions.Two reviewers raised issues relating to whether or not Clp1 3'UTR shortening is a mechanism related to Aire's function or may likewise affect Aire independent genes (See title of the paper). This important point should be fully addressed in a revised manuscript, including via answers to the specific points raised below:– To support the claim that 3'UTR shortening is due to Clp1, the authors are asked to address in more detail whether the Aire-sensitive genes influenced by Aire-KO and Clp1-KD are mostly shared or with the similar pattern around pA sites. If not, why?

We thank the reviewers for highlighting this important question about the relationship between the Aire-sensitive genes and the pA effects of *Clp1* KD.

* First, we would like to reiterate that in our paper, we conclude that Clp1 promotes/ contributes to 3'UTR shortening at Aire-sensitive genes but do not rule out contributions by other factors. Figure 3D Left and Figure 4B show that the levels of d3'UTR ratios of Aire-sensitive genes in *Clp1* KD samples do not reach those of Aire-neutral genes in the Ctr samples. We noted in the text that the data suggested that Clp1 can account for most, but perhaps not all, of the 3’UTR shortening effect, such that other additional factors may also contribute to the increased proportion of short-3'UTRs of Aire-sensitive genes (Results sections “CLP1 promotes 3’UTR shortening and higher expression at Aire-sensitive genes in HEK293 cells” and “Clp1 promotes 3’UTR shortening and higher levels of Aire-upregulated transcripts in mTEChi”).

* Regarding the specificity of the pA effect of Clp1 at Aire-sensitive genes, we assessed the magnitude of the difference in 3’UTR shortening for the Aire-sensitive versus Aire-neutral genes upon *Clp1* KD. We determined the proportion of genes that exceeded a high threshold of short-3’UTR transcript isoform enrichment upon *Clp1* KD, namely d3’UTR ratios of < 0.5 in Ctr vs. *Clp1* KD mTEChi. For the Aire-sensitive genes, 181 out of 528 genes (34%) exhibited this large shift to the shorter 3’UTR, whereas for the Air-neutral genes, a significant smaller proportion, 1206 out of 6076 genes (20%) exhibited this degree of shift to the shorter 3’UTR form (Chisq test: P=9.0*10^-15^). Please see new Figure 4—figure supplement 2 and Results section “Clp1 promotes 3’UTR shortening and higher levels of Aire-upregulated transcripts in mTEChi”.

We also observed an enhanced proportion of genes subject to CLP1-mediated 3'UTR shortening among Aire-sensitive versus Aire-neutral genes in HEK293 cells (for CLP1-sh2: 33% in Aire-sensitive genes versus 9% in Aire-neutral genes, Chisq test: P=9.7*10^-23^; for CLP1-sh3: 28% versus 10%, Chisq test: P=5.7*10^-13^). Please see new Figure 3—figure supplement 3 and Results section “Clp1 promotes 3’UTR shortening and higher levels of Aire-upregulated transcripts in mTEChi”.

Thus, these results strongly support a highly preferential, but not necessarily exclusive 3'UTR shortening effect of Clp1 at Aire-sensitive genes versus other genes. We therefore modified in the Discussion the sentences that may have implied that the effect of Clp1 was strictly specific to Aire-sensitive genes.

* We also now highlight a correlation between Clp1-mediated 3'UTR shortening and increased expression at Aire-sensitive genes. Namely, in WT mTEChi the Aire-sensitive genes responsive to the 3'UTR shortening effect of Clp1 exhibited higher levels of expression than the Aire-sensitive genes neutral to Clp1’s effect. Please see new Figure 4—figure supplement 3 and Results section “Clp1 promotes 3’UTR shortening and higher levels of Aire-upregulated transcripts in mTEChi”.

– The data presented all point towards use of proximal pA sites through the action of Clp1 being an independent layer of control of TRA expression (in the sense of being 'upstream' of Aire's function rather than being a consequence of Aire's action). mTECs are thought to express 'many' TRAs in an Aire independent manner (or do the authors feel otherwise?). Are these also affected by knockdown of Clp1? In sum, the question arises whether 'Aire dependent genes undergo Clp1-mediated 3'UTR shortening….' (as the title insinuates) or whether 'Weakly expressed genes undergo Clp1-mediated 3'UTR shortening…'

We thank the reviewers for calling our attention to this point. Indeed, since Aire-sensitive genes are mainly composed of TRA genes that exhibit similar low levels of expression than Aire-independent TRA genes, we acknowledge that it is important to determine whether the Clp1-mediated 3’UTR shortening was preferentially affecting the Aire-sensitive genes or the TRA genes.

To address this point, we selected, within the whole set of Aire-neutral genes, the genes that have been unambiguously identified as either TRA genes and non-TRA genes (Sansom et al., 2014) and investigated the effect of *Clp1* KD on the d3'UTR ratios of these two gene sets. We found only small statistically insignificant increases of d3'UTR ratios for either the TRA or non-TRA Aire-neutral genes following *Clp1* KD, indicating that the preference of genes for Clp1’s effect is more aligned with Aire-sensitivity than whether the gene is a TRA gene. Please see revised Figure 4B and Results section “Clp1 promotes 3’UTR shortening and higher levels of Aire-upregulated transcripts in mTEChi”.

Two reviewers also asked for further data related to the role of Clp1 in mTECs. Again, these points should be addressed in a revised manuscript.– Please provide additional data on how Clp1-driven 3'UTR shortening can lead to high transcript stability. Are any specific microRNAs or RNA-binding proteins involved during this process?

Short-3’UTR transcripts, in comparison to their longer counterparts, lack miRNA-binding sites and AU-rich elements, which are specifically targeted by miRNAs and RNA-binding proteins (RBPs) respectively. While miRNA targeting is known to destabilize transcripts and decrease their translation, RBP targeting can either trigger repression or activation signals depending on the type of cis elements that they recognize. Hence, transcript stability is controlled by the net effects of miRNAs and RBPs. In our study, we observe a clear association of Clp1-driven 3'UTR shortening with higher transcript stability in mTEChi, strongly suggesting that the 3'UTR-shortened transcripts escape the post-transcriptional repression mediated by potent miRNAs and/or destabilizing RBPs in mTEChi.

mTEChi correspond to a stage of TEC development that directly arises from immature mTEClo. Taking mTEClo as a comparator, we assessed the level of miRNA-mediated post-transcriptional repression in mTEChi versus mTEClo precursor cells by assessing the differences in expression of sets of genes identified as potentially targeted by miRNAs, in mTEChi versus mTEClo. To this end, we performed a Gene Set Enrichment Analysis (GSEA) between the two mTEC populations for each set of genes potentially targeted by a specific miRNA or miRNA family (as defined by the TargetScan database, new Figure 5—source data 1). We found 346 miRNA-specific target gene sets that had significantly reduced expression in mTEChi versus mTEClo. In sharp contrast, we did not detect any of the miRNA-specific target gene sets that had reduced expression in mTEClo versus mTEChi. This suggests that miRNAs are an important contributor to transcript destabilization in mTEChi and that this effect is similar or less for the same miRNA target sets in mTEClo cells. A GSEA leading edge analysis of the 346 miRNA-specific gene sets revealed 11259 target genes that accounted for the gene-set reduction in mTEChi, including 5532 genes targeted by a least 5 different miRNAs. Please see new Figure 5—figure supplement 1, new Figure 5—source data 1, Results section “Clp1-driven 3’UTR shortening of Aire-upregulated transcripts is associated with higher stability in mTEChi” and Materials and methods sections “miRNA potential target gene identification” and “Gene set enrichment analysis (GSEA)”. We updated our GEO depository GSE140815 with RNA-seq data of mTEChi and mTEClo isolated from the same pool of thymi (Materials and methods section “RNA-seq and d3’UTR ratios”).

To determine whether the Clp1-promoted 3'UTR shortening at Aire-sensitive genes can lead to escape from post-transcriptional repression mediated by the miRNAs associated with these 346 target-gene sets, we calculated, among the Aire-sensitive genes subject to Clp1-promoted shortening, the proportion of those that are potentially targeted by one of these miRNAs at a conserved site in the d3’UTR of their longer-transcript isoforms (new Figure 5—source data 1). We found a large proportion of these cases (46%) had one or more such conserved miRNA target sites, providing a specific miRNA de-repression mechanism due to 3’UTR shortening for a large fraction of the Aire-sensitive genes. This analysis is only able to identify miRNAs that are highly conserved and that have disproportionate effect in mTEChi versus mTEClo, so it is expected to miss many of the cases of potential miRNA regulation that may be relieved by 3’UTR shortening in mTEChi. Please see new Figure 5—source data 1 and Results section “Clp1-driven 3’UTR shortening of Aire-upregulated transcripts is associated with higher stability in mTEChi”.

These results strongly suggest that the Aire-induced transcripts that undergo Clp1-driven 3’UTR shortening in mTEChi escape the post-transcriptional repression mediated by a variety of miRNA species, resulting in higher stability of these transcripts in comparison to the longer 3’UTR transcripts.

– Unfortunately, the authors only address the effect of Clp1 knockdown in the absence of Aire in HEK293 cells or in the presence of Aire in mTECs. It would be extremely interesting to relate this to the effect-size of Aire in HEK293 cells and in mTECs (e.g. how would Figure 3D, right, look like for Aire-expressing HEK293 cells in which Clp1 has been knocked-down). Alternatively (but certainly asking too much), it would have been very informative to see how Figure 4B, right, would look like for Aire-KO mTECs).

We thank the reviewers for highlighting this question. To address it, we generated new RNA-seq data, as described below, that we deposited in the GEO database as an update of our depository GSE140738.

We infected HEK293 cells with CLP1-sh2, sh3 or the Ctr shRNA, and for each, performed transfection with the *Aire* expression plasmid. After qPCR validation of *CLP1* knockdown (>50% KD), we generated RNA-seq data for each of these samples and analyzed the level of expression of the Aire-sensitive genes. These experiments revealed a significant effect of *CLP1* knockdown, for both CLP1 shRNAs, on the levels of Aire-sensitive gene expression in the *Aire*-transfected HEK293 cells, similar in magnitude, and apparently slightly greater than in AIRE-negative HEK293 cells. Please see new Figure 3—figure supplement 4 (in comparison to Figure 3D Right) and Results section “CLP1 promotes 3’UTR shortening and higher expression at Aire-sensitive genes in HEK293 cells”.

Bioinformatics: The reviewers raised a number of concerns and questions regarding bioinformatics methods and analyses, which should be fully addressed in a revised manuscript.1) Please provide more detail on how the d3'UTR ratio, used to measure 3'UTR usage, was calculated. Did the method used consider non-uniform read bias? Whether d3'UTR ratio is based on transcript or genes? Also, please explain why d3'UTR ratio has a maximum value of 2 based on Figure 1C, E, Figure 2A etc?

* Non-uniform read coverage can be an obstacle to the comparison of read density between different parts of the transcripts, in particular between the 3' ends and the upstream regions. One source of 3’ biased increased read coverage is mRNA degradation associated with mRNA selection through polyA tail targeting. In order to reduce mRNA degradation as much as possible, we used TRIzol for all RNA extractions, notably TRIzol LS for sorted cells. Importantly, we sequenced only high-quality RNA (with RIN values around or over 8). Please see Materials and methods section “RNA-seq and d3’UTR ratios”.

Another source of 3' biased increased read coverage is the retro-transcription during the preparation of the RNA-seq library if oligo-dTs are used. In this project all RNA-seq libraries were generated using the Truseq (Illumina) or the Smarter (Clontech)/ Nextera (Illumina) chemistry, both including retro-transcription with random primers, therefore avoiding additional bias in 3' read coverage.

To lower the impact of non-uniform RNA-seq coverage between 3' ends and their upstream regions on the calculation of d3'UTR ratios, we normalized the read density of the cleavable d3’UTR region of each gene with a potential pA, by the read density of its constitutive contiguous upstream region in its last 3’ exon, so that the two considered regions used for d3’UTR ratio calculation are in close proximity within 3’ ends of the genes. Please see Materials and methods section “RNA-seq and d3’UTR ratios”.

* We provide more detail about how we calculated distal 3’UTR ratio here and in the revised text. After the mapping of the RNA-seq reads to the mm9 genome using Bowtie, we counted the reads that lie within the d3'UTR regions and within their contiguous upstream regions of the last 3’ exons using intersectBed, coverageBed and a GTF file with d3’UTR annotations that we generated as follows:

First, from the UCSC Table Browser (https://genome.ucsc.edu/cgi-bin/hgTables), we chose mm9 mouse genome/ Genes and Gene Predictions/ RefSeq Genes/ refGene, and we obtained one file with coding exon features only and another file with 5' and 3'UTR features. We combined these two files into a single one and split the 3'UTR annotations into distal 3'UTR (d3'UTR) and proximal 3'UTR annotations at the sites of alternative pAs that we identified in parsing the PolyA_DB 2 database.

Then, we fused into a single feature the proximal 3'UTR annotation with the last coding exon (exon CDS) and formatted the file to fit a GTF format (revised Figure 1—source data 1). For each Refseq gene corresponding to an independent NM_Refseq ID, we divided the RPKM expression of the d3'UTR region by the expression of the upstream region of the last 3’ exon to generate the d3'UTR ratio used as a measure of pA usage. Please see revised Figure 1—source data 1 and Materials and methods section “RNA-seq and d3’UTR ratios”.

* To identify the genes differentially expressed between two conditions, we counted the reads that map to all exons of the genes, normalized the counts by using DESeq and computed the expression foldchange. We used, for read counting, a pre-formatted GTF file obtained from the UCSC Table Browser, in choosing mm9 mouse genome/ Genes and Gene Predictions/ RefSeq Genes/ refGene. Please see new Figure 1—source data 2 and Materials and methods section “RNA-seq and d3’UTR ratios”.

* d3'UTR ratios mostly range from 0 to 2 with a small number of genes with ratios reaching up to 10 – 30. These out of scale values possibly result from hampered sequencing of constitutive 3’ regions upstream of alternative pAs due to local high GC content, or from increased transcription within cleavable distal 3'UTR regions due to the presence of cryptic transcriptional start sites. For clearer representation in Figures 1C, E, Figure 2A, Figure 1—figure supplement 1A, B, we cropped out the negative density values – d3'UTR ratios are positive – and the density values above 2 corresponding to a minority of events. However, to provide a complete picture of the d3'UTR ratio distribution, as well as the most accurate relationship between d3’UTR ratios and levels of expression, we have generated scatterplots including all d3'UTR ratios. Please see new Figure 1—figure supplement 2 and Results section “Aire-sensitive genes show a preference for short-3’UTR transcript isoforms in mTEChi”.

2) Equal number Aire-neutral datasets were used as the control throughout the manuscript. The authors are asked to address why this small arbitrarily selected dataset can represent the global pattern of Aire-neutral genes. They should describe how the Aire-neutral genes were selected, and whether these are equally low in their basal expression levels (see below).

* P-values, including those obtained from the Wilcoxon test, are sensitive to sample size and tend to decrease as n increases. The Aire-neutral genes, with an expression foldchange between 0.5 and 2, correspond to thousands of genes, i.e., a much higher number than the Aire-sensitive genes (Figure 1B). For each comparison, to prevent a biased reduction of the p-values due to the very large size of the Aire-neutral sample, we matched its size to the one of the tested Aire-sensitive sample, in selecting the Aire-neutral genes with the closest-to-1 expression foldchange. For instance, in the WT mTEChi comparison of d3’UTR ratios (Figure 1C), we selected the Aire-neutral genes with an expression FC between 1/1.06 and 1.06 to match the size of the Aire-sensitive sample (n=947). In this selection, the genes "less neutral" to Aire's action, i.e., with expression FC away from 1 and closer to 0.5 or 2 were discarded. In addition, performing the d3’UTR ratio comparison using two different Aire-neutral 947 gene subsamples – one corresponding to expression FC between 1.06 and 1.3 and the other to expression FC between 1/1.18 and 1/1.06 – also led to high statistically significant d3’UTR ratio reduction of Aire-sensitive genes (P=8.3*10^-12^ and 8.1*10^-9^, respectively). Please see Materials and methods section “Statistical analysis”.

* In the Aire-sensitive/neutral gene comparisons, the Aire-neutral genes were not expression-matched to the Aire-sensitive genes. To fully address the reviewers’ concern about basal expression levels, we have taken into account the level of gene expression in performing a local regression (loess) of the d3'UTR ratios across expression levels of Aire-sensitive genes and the whole set of Aire-neutral genes in WT and *Aire*-KO mTEChi. While the d3’UTR ratios vary dramatically across genes, at all expression levels, the loess-fitted curve of the Aire-sensitive genes is significantly lower than the one of the Aire-neutral genes, therefore revealing, in WT and *Aire*-KO mTEChi, a preference of Aire-sensitive genes for smaller d3’UTR ratios that is independent of the levels of gene expression. Please see new Figure 1—figure supplement 2 and Results section “Aire-sensitive genes show a preference for short-3’UTR transcript isoforms in mTEChi”.

3) Aire-sensitive genes are generally expressed at very low levels as compared to most other genes (= Aire-insensitive/neutral genes) (see also Figure 1B). Please comment on whether or not in the case of 'low abundance' mRNAs, there might be a general technical bias towards detection of more 5' reads even among reads that are all upstream of the proximal pA site (which would result in an artificially low d3'UTR ratio). In this context, are the selected Aire neutral genes on average expressed at a similarly low level (see for instance Figure 4B, right: here, the median expression (RPKM) of Aire-neutral genes is substantially higher than that of Aire-sensitive genes; see also Figure 3D)?

In the Aire-sensitive/neutral gene comparisons, the Aire-neutral gene subsamples used as controls have the same average expression than the whole set of Aire-neutral genes. The Aire-sensitive genes are expressed at much lower levels than the Aire-neutral gene subsamples. The local regression of d3’UTR ratios against the levels of expression of Aire-neutral or Aire-sensitive genes (previous point and new Figure 1—figure supplement 2) showed that the low expressed genes exhibit, on average, *higher* d3’UTR ratios than the highly expressed genes, therefore showing that the smaller d3’UTR ratios associated with Aire-sensitive genes do not result from a biological or technical general bias due to their low expression.

4) Related to Figure 3A, the authors are asked to explain why some known APA factors did not change d3'UTR ratio.

The 3’ end processing complex is composed of ~35 core 3’ processing factors, including Clp1, that are organized into four specialized RNA binding sub-complexes, as well as ~50 accessory proteins that interact with the core factors and mediate crosstalk between 3’ processing and other cellular processes (Shi et al., 2009). The effect of a number of core factors on 3’UTR length variation has been evaluated in various cellular systems, showing that many of them have no effect on 3’UTR lengthening or shortening (Li et al., 2015). The absence of effect could indicate that these factors do not contribute to the activity of 3’ end processing sub-complexes or that other factors, perhaps with redundant function, can compensate for their loss. For instance, the lack of 3’UTR length variation following the knockdown of Cstf2, a member of the Cstf sub-complex, has been proposed to result from the redundant function of Cstf2T and Cstf2 (Martin et al., 2012, Li et al., 2015). Please see Results section “CLP1 promotes 3’UTR shortening and higher expression at Aire-sensitive genes in HEK293 cells”.

5) For the multi-tissues analysis from various sources, did the authors consider batch effect and other technical bias? Specifically, related to Figure 1G, an important question is whether a gene that shows a short 3'UTR preference in mTECs also does so in the peripheral tissue. Is that what Figure 1G is supposed to address? It is very difficult to understand. Would it be feasible to plot for all the individual 574 Aire-sensitive TRAs from Figure 1D their d3'UTR ratio in mTEChi vs. their d3'UTR ratio in the respective peripheral tissue to ask in how far the distribution of datapoints deviates from a diagonal? Possibly, this might be a more intuitive way of showing whether or not the usage of a proximal 3' UTR differs between mTEChi and a respective peripheral tissue.

* For the multi-tissue comparison, we included twenty tissue RNA-seq datasets generated by a single lab: that of Bing Ren at UCSD. These datasets were deposited in GEO under two different accession numbers: GSE29278 corresponding to a partial set and GSE36026 to the full set. We modified the Materials and methods section “Multi-tissue comparison analysis” to refer only to the GSE36026 accession number. These RNA-seq datasets were generated as part of the ENCODE project using standardized protocols in order to enable between-sample comparison. In addition to these twenty RNA-seq libraries, the two other libraries that we selected for the multi-tissue comparison, as well as the libraries corresponding to our mTEC samples, we all generated from polyA+ RNA using Illumina chemistry and sequenced on Illumina instruments.

A main artefact of the RNA-seq library preparation is biased duplicate reads that arise from reverse transcription, PCR or RNA fragmentation bias (Roberts et al., 2011; Sendler et al., 2011), and that introduce local distortion in the read coverage of the transcripts. These artefact duplicate reads are not readily distinguishable from the natural ones that arise from over-sequencing of highly expressed genes. Hence, duplicate reads, including the normal and artefact ones, are usually not removed from RNA-seq data, in which they contribute to the high dynamic range of gene expression values. Here, since our main interest focuses on the local read distribution within 3' ends between a series of samples that were not all generated by the same research center, we chose a conservative way to correct for the artefact reads, in removing all duplicate reads. Please see Materials and methods section “Multi-tissue comparison analysis”.

In addition, we have confirmed that small d3'UTR ratios of the Aire-sensitive genes in mTEChi were not correlated with the sequencing depth of the mTEChi RNA-seq libraries nor with the length of the generated reads (50bp). This result was obtained by generating successive subsamples of mapped reads (up to the lowest depth of the tissue RNA-seq libraries) by trimming the reads to 36bp (read length of Bing Ren's libraries), and by looking for an effect on d3’UTR ratios. We observed that the high sequencing depth and the 50bp read length have a very limited effect on d3'UTR ratios and that this very limited effect rather goes towards higher d3'UTR ratios. Please see new Figure 1—figure supplement 3 and Results section “Aire-sensitive genes show a preference for short-3’UTR transcript isoforms in mTEChi”.

* We followed the reviewers’ recommendation and simplified Figure 1G in a way that only the d3'UTR ratios of the Aire-sensitive TRA genes are shown. The preference for short-3'UTR transcript isoforms of these genes in mTEChi resulted from the comparison of the d3'UTR ratios across all tissues and the finding that the d3'UTR ratios of the Aire-sensitive TRA genes in mTEChi are amongst the smallest. Please see revised Figure 1G and Results section “Aire-sensitive genes show a preference for short-3’UTR transcript isoforms in mTEChi”.

We have generated a scatterplot of the d3'UTR ratios of 762 TRA genes (identified in the multi-tissue comparison) in mTEChi versus their respective tissue of expression and found a dramatic bias towards higher ratios in peripheral tissues, confirming the preference of Aire-sensitive TRA genes for short-3’UTR transcript isoforms in mTEChi, as compared to the periphery. Please see revised Figure 1H and the aforementioned section.

Roberts A, Trapnell C, Donaghey J, Rinn JL, Pachter L. 2011. Improving RNA-Seq expression estimates by correcting for fragment bias. Genome Biol 12: R22.

Sendler E, Johnson GD, Krawetz SA. 2011. Local and global factors affecting RNA sequencing analysis. Anal Biochem 419: 317–322.

6) Please ensure that all of the bioinformatics methods and tools used are appropriately referenced, as this is currently not the case.

* We have clarified how we generated the annotation files including the d3’UTR features delineated through proximal (alternative) pA genomic location obtained from the PolyA_DB 2 database. These annotation files used for RNA-seq and microarray analyses, as well as those used for RNA-seq differential expression, are provided in Figure 1—source data 1 and 2. We have also described more precisely how we carried out read count and differential expression and we provide the R-script that we used to calculate d3’UTR ratios in WT and *Aire*-KO mTEChi (new Figure 1—source data 3). References to the softwares that we used were added. Please see new Figure 1—source data 1, 2, 3, Materials and methods section “RNA-seq and d3’UTR ratios” and Results section “Aire-sensitive genes show a preference for short-3’UTR transcript isoforms in mTEChi”.

* We added the reference of the software used for CLIP-seq analysis and described more precisely how this analysis was carried out in using a CLIP-seq wig file and bed files containing proximal and distal pA location of Aire-sensitive or neutral genes. These bed files were generated using files corresponding to proximal and distal pA genomic location of genes extracted from the PolyA_DB 2 database (new Figure 2—source data 1). Please see new Figure 2—source data 1 and Materials and methods section “CLIP-seq analysis”.

* We have clarified how we performed microarray differential expression using the aroma.affymetrix R-package and how we extracted expression values of each individual probe of all genes on the array using the same R-package but with additional options listed in the new Figure 3—source data 1. We have also described how we mapped the individual probes to the d3’UTR features and calculated the d3’UTR ratios in providing our R-script and its dependent files (new Figure 3—source data 2). Finally, we provide the R-script that we used to assay the d3’UTR imbalance and its statistical significance (new Figure 3—source data 3). Please see new Figure 3—source data 1, 2, 3, Materials and methods section “Microarray gene expression profiling”, and “Individual probe-level microarray analyses”, and Results section “CLP1 promotes 3’UTR shortening and higher expression at Aire-sensitive genes in HEK293 cells”.

* We have also described how we used GSEA to test the effect of candidate gene knockdown on the expression of Aire-sensitive genes and the potential involvement of miRNAs or miRNA families in the regulation of sets of genes in mTEChi versus mTEClo. Details, as the list of potential target genes harboring miRNA conserved sites in 3'UTRs generated through parsing the TargetScan 6.2 database, were included (new Figure 5–source data 1). Please see new Figure 5—source data 1 and Materials and methods section “miRNA potential target gene identification” and “Gene set enrichment analysis (GSEA)”.

Textual changes:Concerns were raised that the existing literature is only incompletely discussed, particularly in the Introduction. Please include further discussion of the current literature around post-transcriptional control mechanisms in mTECs involving changes in the mRNA 3´UTRs, and fully cite relevant previous references.

We thank the reviewers for highlighting the need to include the current literature on miRNAs and post-transcriptional regulation in mTECs. This notably echoes with the specific miRNA de-repression mechanism that we highlight in the revised manuscript. Please see the Discussion section.

Also, subsection “Aire-sensitive genes show a preference for short-3’UTR transcript isoforms in mTEChi and in some peripheral tissues”, last paragraph, first sentence: This somewhat contrived sentence may deserve attention.

We edited the sentence to make it smoother and more meaningful. Please see Results section “Aire-sensitive genes show a preference for short-3’UTR transcript isoforms in mTEChi”.